



# ET-WB: water balance-based estimations of terrestrial evaporation over global land and major global basins

Jinghua Xiong[1][#]; Abhishek[2][#]; Li Xu[3]; Hrishikesh A. Chandanpurkar[3]; James S. Famiglietti[3,4]; Chong Zhang[5]; Gionata Ghiggi[6]; Shenglian Guo[1]; Yun Pan[5]; Bramha Dutt Vishwakarma[7,8]

[1]State Key Laboratory of Water Resources and Hydropower Engineering Science, Wuhan University, Wuhan, 430072, Hubei, China
[2]Department of Civil Engineering, Indian Institute of Technology Roorkee, Roorkee 247667, India
[3]University Saskatchewan, Global Institute for Water Security, Saskatoon, SK S7N 3H5, Canada
[4]Arizona State University, Global Futures Laboratory, Tempe, AZ, 85281, United States
[5]Beijing Laboratory of Water Resources Security, Capital Normal University, Beijing, 100048, China
[6]Environmental Remote Sensing Laboratory (LTE), EPFL, 1005 Lausanne, Switzerland
[7]Interdisciplinary Centre for Water Research, Indian Institute of Science, Bengaluru, 560012, India
[8]Centre for Earth Sciences, Indian Institute of Science, Bengaluru, 560012, India

[#]*Jinghua Xiong and Abhishek contributed equally.*

*Correspondence to*: Shenglian Guo (slguo@whu.edu.cn); Yun Pan (pan@cnu.edu.cn)

## Abstract

Evaporation (ET) is one of the crucial components of the water cycle, which serves as the nexus between global water, energy, and carbon cycles. Accurate quantification of ET is, therefore, pivotal in understanding various earth system processes and subsequent societal applications. The prevailing approaches for ET retrievals are either limited in spatiotemporal coverage or largely influenced by choice of input data or simplified model physics, or a combination thereof. Here, using an independent mass conservation approach, we develop water balance-based ET datasets (ET-WB) for the global land and the selected 168 major river basins. We generate 4669 probabilistic unique combinations of the ET-WB leveraging multi-source datasets (23 precipitation, 29 runoff, and 7 storage change datasets) from satellite products, in-situ measurements, reanalysis, and hydrological simulations. We compare our results with the four auxiliary global ET datasets and previous regional studies, followed by a rigorous discussion of the uncertainties, their possible sources, and potential ways to constrain them. The seasonal cycle of global ET-WB possesses a unimodal distribution with the highest (median value: 65.61 mm/month) and lowest (median value: 36.11 mm/month) values in July and January, respectively, with the spread range of roughly ±10 mm/month from different subsets of the ensemble. Auxiliary ET products illustrate similar intra-annual characteristics with some over/under-estimation, which are completely within the range of the ET-WB ensemble. We found a gradual increase in global ET-WB from 2003 to 2010 and a subsequent decrease during 2010-2015, followed by a sharper reduction in the remaining years primarily attributed to the varying precipitation. Multiple statistical metrics show reasonably good accuracy





of monthly ET-WB (e.g., a relative bias of ±20%) in most river basins, which ameliorates at annual scales. The long-term
mean annual ET-WB varies within 500-600 mm/yr and is consistent with the for auxiliary ET products (543-569 mm/yr).
Observed trend estimates, though regionally divergent, are evidence of the increasing ET in a warming climate. The current
dataset will likely be useful for several scientific assessments centering around water resources management to benefit society
at large. The dataset is publicly available at https://doi.org/10.5281/zenodo.7314920 (Xiong et al., 2023).

## 1 Introduction

Land evaporation (ET), the total amount of water evaporating from the land surface to the atmosphere, is a crucial component
of the terrestrial water cycle (Rodell et al., 2015; Wang and Dickinson, 2012). It includes the water evaporating from the bare
soil, open water bodies, canopy-intercepted precipitation, sublimation, and transpiration from the plant stomata (Miralles et
al., 2020). Since the global ET returns about two-thirds of the land precipitation back to the atmosphere, it sustains the water
cycle by providing the moisture supply for precipitation and directly affects the partitioning of the Earth's surface heat fluxes
and subsequent heating and cooling effects (Good et al., 2015; Koster et al., 2004; Oki and Kanae, 2006). Thus, ET links the
Earth's surface and the atmosphere and acts as the key element for the interconnected global water, energy, and carbon cycles
(Jung et al., 2010). Accurate quantification of ET is, therefore, imperative for studying the water cycle changes, freshwater
availability and demand, weather and climate dynamics, earth system processes, and surface energy budget closures. However,
ET is poorly constrained, especially at large scales compared to the other components of the water cycle (Syed et al., 2010;
Jasechko et al., 2013; Chandanpurkar et al., 2017), which may become more uncertain with an intensifying hydrological cycle
under a warming climate. To this end, the trends and variability of the global ET fluxes still remain contested (Dong and Dai,
2017; Fisher et al., 2017; Pascolini-Campbell et al., 2020).

Over the past few decades, ET-based science has advanced significantly across scales from leaf to global scales

(Fisher et al., 2017). Several ET products derived from the data-driven and data assimilation methods, satellite observations,
and simulations from the physically or empirically based land surface models have been developed (Long et al., 2014; Liu et
al., 2016); a community effort that is still ongoing (Miralles et al., 2016). These ET products are dedicated to minimizing the
existing shortcomings stemming from varying spatiotemporal scales and are tailored to specific forcing variables (Miralles et
al., 2016). For example, Moderate Resolution Imaging Spectroradiometer (MODIS) ET data provides regular 1 km$^2$ land
surface ET over 109.03 million km$^2$ of global vegetated land areas at 8-day, monthly and annual intervals (Mu et al., 2011).
Also, recent deep learning-based methods have shown an enhanced ability for global ET estimation when compared against
proxy estimates from satellite observations and sparse in-situ data (Koppa et al., 2022). Despite the large spatial and temporal
scale ET retrievals, all of these datasets inherently possess several uncertainties originating either from the forcing datasets or
propagated uncertainty through the varying model structures or a combination thereof. For example, accurate estimations of
ET utilizing the land surface temperature (LST) or other satellite optical and thermal observations need clear skies and hence





are limited in temporal coverage due to the cloud cover issues (Long et al., 2014; Wang and Dickinson, 2012; Yang and Shang,
2013). Similarly, the mismatch between the spatial scales of the forcing data and the vegetation data, in the case of the
Normalized Difference Vegetation Index-based ET products, can result in large uncertainties (Yang et al., 2013).
Owing to all these uncertainties associated with the different methodological approaches, model assumptions, and
scaling issues, the resulting observed ET estimates and their future projections have huge variations from product to product
(Liu et al., 2016; Wang and Dickinson, 2012; Wang et al., 2015). Such disparities generally impede selecting the most
appropriate ET data and even make it contentious, at times, for their application in various hydrometeorological modeling
studies, management, or policymaking frameworks, among others. Moreover, the traditional estimations and the standards for
the validation of ET solely from ground-based measurements from, for example, lysimeters and eddy covariance flux towers,
are also insufficient for larger basin-scale evaluations because of the sparsely distributed network (Pascolini-Campbell et al.,
2020; Wang and Dickinson, 2012). Such limited point observations can further lead to high spatiotemporal heterogeneity
variability in the ET, suffering mainly from the uncertainties arising from the data gap filling and upscaling beyond their
representative local areas (Liu et al., 2016; Pascolini-Campbell et al., 2020). Therefore, in the context of a changing climate
and continually intensifying human activities, the paramount importance of ET in global and regional water cycles and
associated land-atmosphere interactions fosters the need and underscores the importance of independent, large-scale, and
better-constrained ET estimates.
Since the multifaceted variable, ET, is difficult to measure from space or from in-situ records directly, it has to be
derived through the physically driven models incorporating a variety of controlling atmospheric, radiative, and vegetative
factors (Fisher et al., 2017). However, the recent advancement in mapping the other components of the water cycle, changes
in the terrestrial water storage (TWS), in particular, has enabled an alternate assessment of ET at large basin scales, which
often is the scale of interest in water resources management (Pascolini-Campbell et al., 2020). The Gravity Recovery And
Climate Experiment (GRACE) and its successor GRACE Follow-On (both jointly referred to as GRACE hereafter) have
provided the TWS (sum of all of the water storage components within a land mass) variations with unprecedented accuracy
since 2002 (Tapley et al., 2004; Sneeuw et al., 2014; Rodell et al., 2018). When used in combination with the precipitation and
runoff in a water balance equation, the changes in TWS can be used for an independent and mass conservation-based estimate
of ET, which will be free from most of the above-mentioned shortcomings in the modeled, upscaled, or in-situ products (Rodell
et al., 2004; Bhattarai et al., 2019). Moreover, the resulting ET will be better constrained since the GRACE inferred TWS
contains the embedded signals of both the natural variability and the anthropogenic influences. The major limitation with
GRACE TWS variations is, however, its coarse spatial resolution (Ramillien et al., 2006) which we take the edge off by
limiting our analysis to the global land and major global basins.
Previous studies employing the water balance approach either rely on single datasets of precipitation and/or runoff
(Gibson et al., 2019; Liu et al., 2016) are focussed on the regional scales (Castle et al., 2016; Pascolini-Campbell et al., 2020;



Rodell et al., 2004; 2011; Swann et al., 2017; Wan et al., 2015). A few global studies (e.g., Liu et al., 2016; Miralles et al.,
2016; Ramillien et al., 2006; Zeng et al., 2012; Lehmann et al., 2022) are limited either in terms of data used or in the temporal
coverage. Here, we leverage a multitude of precipitation, runoff, and TWS changes (23, 29, and 7, respectively) datasets and
employ the water balance approach to generate a total of 4669 subsets of ET during 2002-2021 for global land and major 168
river basins. We rigorously assess the uncertainty bounds of the resulting ET and also analyze the relationship with various
attributes such as the basin area, climate (aridity index, AI), and human interventions (irrigation). This water balance approach
checks global and basin scale ET given the spatial accumulation of errors in LSM- or RS-based ET products (Pascolini-
Campbell et al., 2020). Given the ongoing controversy over the reliability of existing ET products, while in situ observation
data are scarce (Douville et al., 2013; Zhang et al., 2016), the inter-comparison of mass-balance derived monthly ET ensemble
estimates with several existing ET datasets provides a way to benchmark and improve the estimate of ET. We expect our
product will be relevant for various scientific and societal applications, including the study of extreme events, water and carbon
cycle, agricultural management, sea level budgeting, biodiversity assessments, global and regional hydrological cycle, water
resources management, ecosystem resilience, and for improving weather predictions across scales (Fisher et al., 2017).

## 2 Methods

### 2.1 Water balance equation

The terrestrial water balance method was used to produce the ET-WB dataset. For a basin scale, it can be written as follows:

$$ET = P - \Delta S - R \pm WD \tag{1}$$

where P is the basin-averaged precipitation, and R is the river flow or runoff going outside the basin. $\Delta S$ is the monthly storage
change which is calculated as the backward difference of the terrestrial water storage (i.e., the changes in the month of
calculation and the previous month), while different computation methods, such as the backward difference combined with a
three-month running average might produce subtle difference (Long et al., 2014; Pascolini-Campbell et al., 2020). WD denotes
the diverted water volume inside/outside the basin. All the water fluxes are on the monthly scale from May 2002 to December
2021 and expressed in the unit of millimeters (mm/month) of equivalent water depth. WD is not considered in our study
because the amplitude of the transferred water of most projects is generally small relative to other water components and/or
directly flows outside the basin through the river channels. Therefore, the WD influences on the water balance ET estimations
might be considered small, even for the 14 major existing projects located across the 168 studied basins from the Global Water
Transfer Megaprojects depository (Shumilova et al., 2018) (Table S1). Although the terrestrial water balance method has been
extensively applied in different river basins of the world (Rodell et al., 2004; Long et al., 2014; Li et al., 2019), a global
database is still lacking, and the systematic uncertainty, variation, and distribution also remain unexplored from a global
perspective.
We performed the calculation over the 168 major river basins of the world from the Global Runoff Data Centre
(GRDC, https://www.bafg.de/GRDC/EN/Home/homepage_node.html) and the global land excluding Antarctic and Greenland
(Fig. 1). These selected basins cover a wide range of climate conditions and human intervention with a minimum area of
~64,000 km$^2$, which is sufficiently large for the retrieval of TWSA from GRACE solutions at basin scale at least in the
hydrology community (Vishwakarma et al., 2018). Apart from the terrestrial water balance, the atmospheric water balance
also offers an effective alternative framework to estimate ET as it is also an important factor in the atmospheric water cycle,
i.e., the residual precipitation, the horizontal divergence of the vapor flux, and the change in column water vapor. Although
such an alternative estimation of ET from the independent atmospheric data can potentially supplement the water balance-
based ET (referred to as 'ET-WB' hereafter), this is outside the scope of our study.

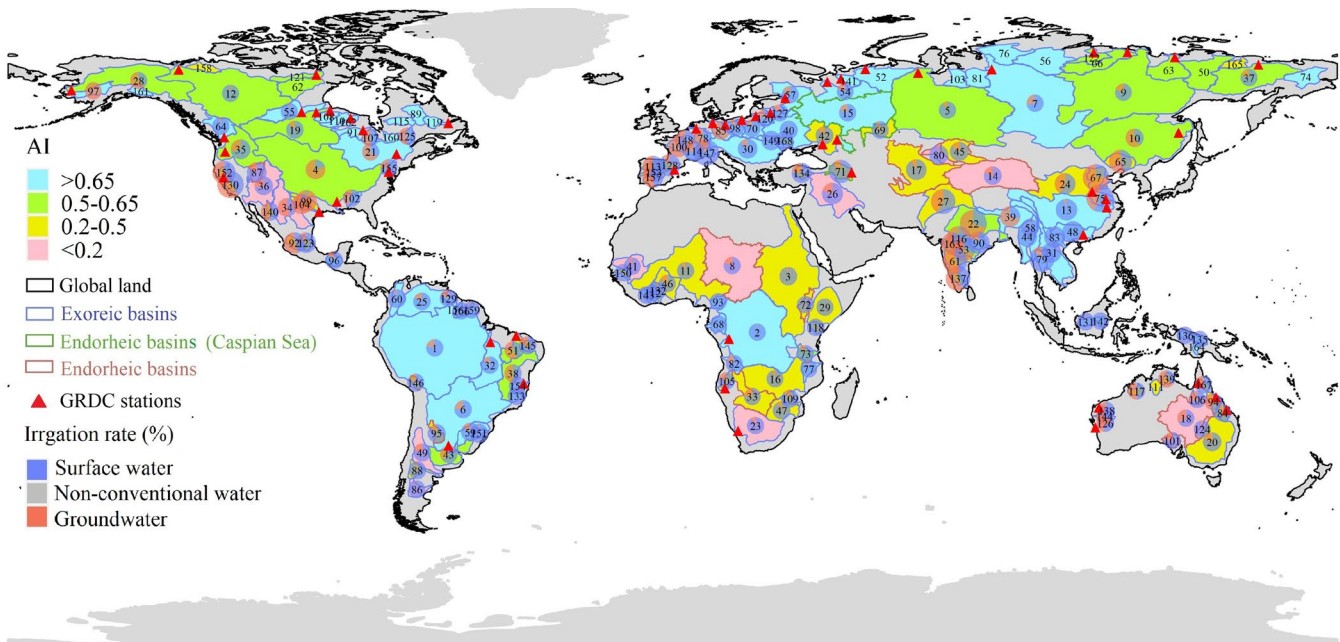

**Figure 1: Location and attributes of the 168 studied river basins. The labeled numbers represent the basin ID. Please find further**
**details in Table S2. The irrigation information is obtained from the latest version of the Food and Agricultural Organization (FAO)**
**Global Map of Irrigated Areas (https://www.fao.org/aquastat/en/geospatial-information/global-maps-irrigated-areas/latest-**
**version/). The aridity index information is collected from the Version 3 of the Global Aridity Index and Potential Evapotranspiration**
**Database (Zomer et al., 2022). The inserted pie chart indicates the percentage of irrigation area from different water sources to the**
**basin area. The radii are proportional to the total percentage of the equipped irrigation area, which has been re-scaled using the**
**natural logarithms after adding 10 to avoid negative (very small) values for better visualisation.**
**2.2 Evaluation metrics**
The ET-WB dataset was compared with multiple global ET products (see details in the Data section) at various temporal and
spatial scales. Firstly, the comparisons were conducted at the monthly and annual time scales over global land and selected
river basins to investigate the sensitivity of the ET-WB performance using various evaluation metrics, including Pearson



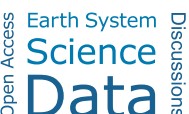

correlation coefficient (CC), Nash-Sutcliffe efficiency (NSE), root mean square error (RMSE), and relative bias (RB). They
describe different aspects of ET-WB performance; for example, CC [-1,1] measures the linear correlation with auxiliary ET
products, and NSE (≤1) determines the relative magnitude of residuals between observations and predictions relative to the
variance of the former. RMSE (⩾0) quantifies the differences between ET-WB and other existing ET products, while it is not
normalized and challenging to compare basins with different ET amplitudes. As such, the metric RB (can be negative or
positive) is used to express the relative bias of ET-WB compared with other ET datasets over the period. Mathematically, these
metrics are defined as follows:

$$CC = \frac{\sum(ET_G - \overline{ET_G}) \cdot (ET_{WB} - \overline{ET_{WB}})}{\sqrt{\sum(ET_G - \overline{ET_G})^2} \cdot \sqrt{\sum(ET_{WB} - \overline{ET_{WB}})^2}} \tag{2}$$

$$NSE = 1 - \frac{\sum(ET_G - ET_{WB})^2}{\sum(ET_G - \overline{ET_G})^2} \tag{3}$$

$$RMSE = \sqrt{\left(\frac{\sum(ET_G - ET_{WB})^2}{n}\right)} \tag{4}$$

$$RB = \frac{\sum(ET_{WB} - ET_G)}{\sum ET_G} \cdot 100\% \tag{5}$$

where $ET_G$ represents the auxiliary global ET products for comparison with the ET-WB, i.e., $ET_{WB}$ in Equations 2-5. Secondly,
further comparisons were performed at the level of long-term mean and trend, which were calculated using Sen's slope method
(Sen, 1968). Sen's slope method can overcome the impacts of outliers on time series and can be more accurate than the
traditional linear regression, especially for the heteroskedastic time series (Sen, 1968). Different temporal coverage of the
auxiliary global ET datasets is considered, so only consistent periods with the ET-WB are used for calculations.

**2.3 Uncertainty estimation**

Uncertainty in ET-WB and its contributing variables (e.g., P) is quantified using different methods. Specifically, we estimated
the uncertainty in various TWSA datasets from GRACE solutions and GHM as the residual after removing the long-term trend,
interannual signals, and seasonal cycles based on the Seasonal and Trend decomposition using Loess (STL) method (Cleveland
et al., 1990). The STL method can robustly decompose the TWSA monthly time series into long-term, seasonal, and residual
components, in which the long-term signal can be further separated as a long-term trend and the non-linear (interannual) signals
(Cleveland et al., 1990; Scanlon et al., 2018; Vishwakarma et al., 2021) as:

$$S_{total} = S_{long-term} + S_{seasonal} + S_{residual} \tag{6}$$

where $S_{total}$ is the original TWSA time series, $S_{long-term}$ is the long-term components of time series consisting of the long-
term trend and the remaining interannual components, $S_{seasonal}$ is the seasonal cycle time series of TWSA, and $S_{residual}$ is
the noise and/or other high-frequency (i.e., sub-seasonal) signals. Further, the uncertainty in ΔS was computed from the



uncertainties in TWSA for back and forward months added in quadrature, followed by the determinations of the root mean
squares (RMS) from different results (Long et al., 2014). However, a few studies also indicate that this method might
overestimate the actual uncertainty as the residual temporal signals might contain real information (e.g., sub-seasonal signals)
(Scanlon et al., 2018). For other water components, including P and R, we assumed the standard deviation (SD) across the
ensemble as the uncertainties since we do not have the formal error budget for the multi-source global products from models,
satellites, and field monitoring networks. Uncertainty in the auxiliary ET products used for comparison with ET-WB is also
estimated using the SD method. It should be noted that the SD estimations may underestimate the actual uncertainty because
of the inadequate number of datasets considered in our study. We took different strategies to estimate the uncertainty in ΔS
and other variables because of the strong correlation of the selected GRACE solutions, which can lead to a very low SD among
datasets. A similar situation can occur in R, where 23 out of 29 R datasets are from the G-RUN ensemble with similar
algorithms (but with different meteorological forcing data). The SD of different auxiliary global ET products was also
calculated for comparison, which can be written as:
$$SD = \sqrt{\frac{\sum(X - \bar{X})^2}{n}} \tag{7}$$

where $X$ is the hydrological time series of different variables. Thus, we could estimate the uncertainty in the ET-WB by
propagating the above uncertainties in quadrature with the assumption of independence and normal distribution among
different water fluxes (Rodell et al., 2004):
$$U_{ET-WB} = \sqrt{\sum \left(U_P{}^2 + U_R{}^2 + U_{\Delta S}{}^2\right)^2} \tag{8}$$

where $U_P$, $U_R$, and $U_{\Delta S}$ are the estimated uncertainty for P, R, and ΔS on the monthly scale, respectively. We utilized the RMS
to         represent         the         average         uncertainty         over         the         whole         study         period         as:
$$RMS = \sqrt{\left(\frac{\sum Y^2}{n}\right)} \tag{9}$$

where $Y$ denotes the monthly estimates of uncertainty in different variables (e.g., ET-WB). The relationships between
uncertainty in ET-WB and basin area, climate condition (aridity), and human activities (irrigation) are also detected to
thoroughly investigate the influential factors on the performance of ET-WB.
**3 Data**
Several criteria are applied to select the appropriate datasets for the development of ET-WB: (1) only the publicly available
global datasets are chosen to increase the transparency and reproducibility of our study, (2) the temporal resolution should be
equal to or smaller than one month, spanning at least from 2002 to 2014, (3) the spatial resolution should be finer than 2° to
constrain the uncertainties over small river basins (~64,000 km² for the minimum), and the spatial coverage should be



(quasi-)global to reach most river basins. Alternative factors like the frequency of data updates (mostly are near-real-time and
a few are yearly), the recognition in the community (some datasets not being widely used were excluded), and the data types
(try taking more categories of datasets into account, e.g., satellite, modeling, reanalysis, and in-situ-based products) are also
considered. As such, we used 23 P, 29 R, and 7 ΔS datasets to generate a total of 4669 subsets of ET-WB during May 2002-
December 2021 over 168 river basins and global land, excluding Greenland and Antarctica. We simultaneously selected the
datasets belonging to the same series but with different versions, for example, GLDAS-v1/v2 and NCER/CFSR, because the
older version (e.g., NCER/NCEP) is still updating, and the improvements of the newer version might not be significant and
consistent over all the regions of the world (Qi et al., 2018, 2020). Despite this, it is acknowledged that it is impossible to
consider all the existing datasets meeting the above inclusion criteria because the development of global datasets is advancing
rapidly. All the selected datasets are provided on a grid cell scale and converted into basin-scale based on the changing area of
grid cells over latitude. Hence the varying spatial resolutions of datasets do not require the up/down-scaling processes in our
study. Moreover, most of the products are on a monthly time scale, consistent with the ET-WB estimations. A few daily
datasets are aggregated into monthly time scales by taking the sum from the first to the last day of the certain month, which
might cause some discrepancies with the GRACE solutions because the time sampling of GRACE products is not strictly
distributed within a month (Tapley et al., 2004). As different datasets might have varying temporal and spatial coverage (Fig.
2), the missing months in recent one or two years due to update latency, as well as the basins suffering from incomplete spatial
coverage, are set as NA values. Please find detailed information on the datasets used in our study in Table 1. A more intuitive
work chain for the generation of ET-WB and the related data processing flow is presented in Fig.2.







**Figure 2: Flowchart and the characteristics of the data sets in the study. Please see the data section for detailed descriptions of the various datasets.**

**Table 1. Datasets used in our study.**

| Variable | Dataset | Type | Reference | Selected period | Temporal resolution | Spatial resolution (longitude× latitude) | Spatial coverage |
|---|---|---|---|---|---|---|---|
| | | | | | | | |





|  |  |  |  |  |  |  |
|---|---|---|---|---|---|---|
|  | G-RUN Ensemble | In-situ based | Ghiggi et al., 2021 | 2002.5-2019.12 | Monthly | 0.5°×0.5° | Global land excluding Antarctica |
|  | LORA-v1.0 | Combined product | Hobeichi et al., 2019 | 2002.5-2012.12 | Monthly | 0.5°×0.5° | Global land excluding Greenland and Antarctica |
|  | WGHM | GHM | Schmied et al., 2021 | 2002.5-2016.12 | Monthly | 0.5°×0.5° | Global land excluding Antarctica |
| Runoff | Global River Flow and Continental Discharge Dataset | In situ | Dai and Trenberth, 2002 | 2002.5-2018.12 | Monthly | Gauge stations | Global major river basins |
|  | GloFAS-v2.1 | Reanalysis | Harrigan et al., 2020 | 2002.5-2021.12 | Daily | 0.1°×0.1° | Global land excluding Antarctica |
|  | GloFAS-v3.0 | Reanalysis | Alfieri et al., 2020 | 2002.5-2018.12 | Daily | 0.1°×0.1° | Global land excluding Antarctica |
|  | GloFAS-v3.1 | Reanalysis | Harrigan et al., 2020 | 2002.5-2021.12 | Daily | 0.1°×0.1° | Global land excluding Antarctica |
|  | ERA5-land | Reanalysis | Muñoz-Sabater et al., 2021 | 2002.5-2021.12 | Monthly | 0.1°×0.1° | Global land |
|  | ERA5 | Reanalysis | Hersbach et al., 2020 | 2002.5-2021.12 | Monthly | 0.25°×0.25° | Global land and ocean |
|  | NOAA CIRES 20th Century-v3 | Reanalysis | Slivinski et al., 2019 | 2002.5-2015.12 | Monthly | 0.702°×0.702° | Global land and ocean |
| Precipitation | JRA55 | Reanalysis | Kobayashi et al., 2015 | 2002.5-2021.12 | Monthly | 55 km×55 km | Global land and ocean |
|  | MERRA2 | Reanalysis | Gelaro et al., 2017 | 2002.5-2021.12 | Monthly | 0.625°×0.5° | Global land excluding Greenland and Antarctica |
|  | NCEP NCAR-Reanalysis 1 | Reanalysis | Kistler et al., 2001 | 2002.5-2021.12 | Monthly | 1.875°×1.9048° | Global land and ocean |



| | | | | | | |
|---|---|---|---|---|---|---|
| NCEP DOE-Reanalysis 2 | Reanalysis | Kanamitsu et al., 2002 | 2002.5-2021.12 | Monthly | 1.875°×1.9048° | Global land and ocean |
| CFSR-v1&2 | Reanalysis | Saha et al., 2010 | 2002.5-2021.12 | Monthly | 0.5°×0.5° | Global land and ocean |
| WFDEI | Reanalysis | Weedon et al., 2014 | 2002.5-2016.12 | Monthly | 0.5°×0.5° | Global land excluding Antarctica |
| PERSIANN CDR-v1 | Satellite | Ashouri et al., 2015 | 2002.5-2021.12 | Daily | 0.25°×0.25° | 60° S–60°N |
| TRMM 3B43-v7 | Satellite | Huffman et al., 2007 | 2002.5-2019.12 | Monthly | 0.25°×0.25° | 50° S–50°N |
| GSMaP | Satellite | Okamoto et al., 2005 | 2002.5-2021.12 | Monthly | 0.1°×0.1° | 60° S–60°N |
| CHIRPS-v2.0 | Satellite | Funk et al., 2015 | 2002.5-2021.12 | Daily | 0.25°×0.25° | Global land between 50° S–50°N |
| GPM IMERG-v06 | Satellite | Huffman et al., 2019 | 2002.5-2021.9 | Monthly | 0.1°×0.1° | 60° S–60°N |
| GPCP-v3.2 | Satellite | Huffman et al., 2022 | 2002.5-2020.12 | Monthly | 0.5°×0.5° | Global land and ocean |
| CRU TS-v4.06 | In situ-based | Harris et al., 2020 | 2002.5-2021.12 | Monthly | 0.5°×0.5° | Global land excluding Antarctica |
| GPCC-v2020 | In situ-based | Schneider et al., 2020 | 2002.5-2019.12 | Monthly | 0.25°×0.25° | Global land excluding Antarctica |
| CPC Unified | In situ-based | Chen and Xie, 2008 | 2002.5-2021.12 | Daily | 0.5°×0.5° | Global land |
| MSWEP-v2.8 | Combined product | Beck et al., 2019 | 2002.5-2021.12 | Monthly | 0.1°×0.1° | Global land and ocean |
| PGF-v3 | Combined product | Sheffield et al., 2006 | 2002.5-2016.12 | Monthly | 0.25°×0.25° | 60° S–90°N |
| GLDAS-v1 | Combined product | Rodell et al., 2004 | 2002.5-2019.6 | Monthly | 1°×1° | Global land excluding Antarctica |





| | GLDAS-v2.0 | Combined product | Rodell et al., 2004 | 2002.5-2014.12 | Monthly | 1°×1° | Global land excluding Antarctica |
|---|---|---|---|---|---|---|---|
| | GLDAS-v2.1 | Combined product | Rodell et al., 2004 | 2002.5-2021.12 | Monthly | 1°×1° | Global land excluding Antarctica |
| Actual evaporation | MODIS16 | Satellite | Mu et al., 2011 | 2002.5-2014.12 | Monthly | 0.5°×0.5° | 60° S–80°N |
| | FLUXCOM | In situ-based | Jung et al., 2019 | 2002.5-2015.12 | Monthly | 0.5°×0.5° | Global land excluding Antarctica |
| | GLEAM-v3.6a | Satellite | Martens et al., 2017 | 2002.5-2021.12 | Monthly | 0.25°×0.25° | Global land |
| | WGHM | GHM | Schmied et al., 2021 | 2002.5-2016.12 | Monthly | 0.5°×0.5° | Global land excluding Antarctica |
| Terrestrial water storage anomaly | GRACE CSR RL06 mascons-v02 | Satellite | Save et al., 2016 | 2002.4-2021.12 | Monthly | 0.25°×0.25° | Global land and ocean |
| | GRACE JPL RL06 mascons-v02 | Satellite | Wiese et al., 2018 | 2002.4-2021.12 | Monthly | 0.5°×0.5° | Global land and ocean |
| | GRACE GSFC RL06 mascons-v02 | Satellite | Loomis et al., 2019 | 2002.4-2021.12 | Monthly | 0.5°×0.5° | Global land and ocean |
| | GRACE CSR RL06 Level-2 SH | Satellite | Swenson and Wahr, 2006 | 2002.4-2021.12 | Monthly | 1°×1° | Global land and ocean |
| | GRACE JPL RL06 Level-2 SH | Satellite | Swenson and Wahr, 2006 | 2002.4-2021.12 | Monthly | 1°×1° | Global land and ocean |
| | GRACE GFZ RL06 Level-2 SH | Satellite | Swenson and Wahr, 2006 | 2002.4-2021.12 | Monthly | 1°×1° | Global land and ocean |
| | WGHM | GHM | Schmied et al., 2021 | 2002.4-2016.12 | Monthly | 0.5°×0.5° | Global land excluding Antarctica |



## 3.1 Precipitation

As summarized in Table 1, 23 precipitation data sets from different sources were used as input for the water balance equation (Eq. 1). Three global datasets based on in-situ observations are collected, including the Climatic Research Unit Time Series (CRU TS) database, the Global Precipitation Climatology Centre (GPCC) project, and the unified suite from NOAA Climate Prediction Center (CPC Unified). They generally rely on the point-scale collections of rain gauges worldwide to interpolate the gridded global products. Specifically, the CRU TS dataset incorporates more than 10,000 gauge stations to derive the monthly global gridded data since 1901 based on the angular-distance weighting method with an annual update (Harris et al., 2020). The GPCC project contains the quality-controlled gauge measurements from approximately 67,200 stations worldwide with at least 10 uninterrupted years of available data and then interpolates and superimposes them on the final gridded product in the corresponding resolution (Schneider et al., 2020). The CRU TS and GPCC datasets have almost identical temporal coverage and resolution and mainly rely on national meteorological agencies and related international institutions like WMO and FAO. The CPC Unified dataset is constructed from over 30,000 rain gauges from Global Telecommunication System (GTS), Cooperative Observer Network (COOP), and other national and international institutions. The daily analysis is released on multiple spatial resolutions over the global domain from 1979 to the present (Chen and Xie, 2008). The main advantages of these gauge-based global datasets stem from their large historical records dating back to the beginning of the 20$^{th}$ century, high accuracy, and effective construction cost. However, they heavily suffer from inhomogeneous spatial distribution and substantial maintenance efforts, especially in developing regions with complicated topography like North Africa and Qinghai-Tibetan Plateau. Therefore, the remote sensing technique has become a popular choice in learning global precipitation information in recent decades, which greatly improves precipitation measurement in ungauged and poorly gauged areas.

Six remote sensing products have been collected to enrich our study, namely the Integrated Multi-Satellite Retrievals (IMERG) for Global Precipitation Measurement (GPM), Global Precipitation Climatology Project (GPCP), Precipitation Estimation from Remotely Sensed Information using Artificial Neural Network-Climate Data Record (PERSIANN-CDR), Tropical Rainfall Measuring Mission with 3B43 algorithm (TRMM 3B43), Global Satellite Mapping of Precipitation (GSMaP), and Climate Hazards Group InfraRed Precipitation with Station data (CHIRPS). The TRMM 3B43 product algorithmically merges the microwave observations from multiple sensors, including precipitation radar and visible and infrared scanner (VIRS) loaded in the TRMM, which is a joint space satellite between NASA and Japan's National Space Development Agency to monitor tropical and subtropical precipitation from 1997 to 2015 (Huffman et al., 2007). Then, the successor GPM mission, an international network of satellites carrying the first space-borne Ku/Ka-band Dual-frequency Precipitation Radar (DPR) and a multi-channel GPM Microwave Imager (GMI), continued to provide the global precipitation data up to the present (Huffman et al., 2019). The IMERG algorithm can integrate all information from satellites constellation at a given time to estimate precipitation on the Earth's surface. The satellite observations in the TRMM era were also re-processed using the IMERG algorithm to create long-term continuous records, but the production stopped at the end of 2019. The GSMaP is a blended satellite-based precipitation dataset from the passive microwave sensors in low Earth orbit and infrared radiometers





in geostationary Earth orbit, which was developed by Japan Aerospace Exploration Agency (JAXA) and became the Japanese
GPM standard product (Okamoto et al., 2005). The GSMaP product can distribute the global precipitation over the region from
60° N to 60° S at a high spatial resolution of 0.1°×0.1°. In addition, the CHIRPS dataset, building on the 'smart' interpolation
techniques and high resolution, long period of precipitation records from the infrared Cold Cloud Duration measurements, is
developed by the USGS and Climate Hazards Group at the University of California. It has supplied precipitation estimates
over global land within the range of 50° N to 50° S since 1981 (Funk et al., 2015). The PERSIANN product applies the trained
artificial neural network on GridSat-B1 infrared satellite data of brightness temperature of cold cloud pixels to produce the
rain rate estimates in the latitude band 60° S-60° N from 1983 to the (delayed) present (Ashouri et al., 2015). The GPCP
precipitation dataset dynamically merges various satellite-based information, such as passive microwave and infrared data,
along with the GPCC gauge measurements, contributing to the monthly precipitation estimates from 1979-present worldwide
(Huffman et al., 2022). To control the systematic bias of the satellite sensors, bias correction based on gauge observations (e.g.,
GPCC) and satellite observations (e.g., GPCP) is necessary, particularly over regions having poor gauge coverage, like Africa
and the ocean.

Although the remote sensing technique is a robust option for global precipitation estimations, it still has some
drawbacks, like the relatively short lifetime, the complexity of the retrieval algorithm, and the need for in-situ observations for
bias correction. Thus, global reanalysis products that synthesize multiple geophysical and climatological data to produce high-
resolution precipitation simulations have been developed. We obtained nine reanalysis datasets, including the fifth-generation
reanalysis product of the European Centre for Medium-Range Weather Forecasts (ERA5), the land component of ERA5
(ERA5-land), the Twentieth Century Reanalysis by NOAA, the University of Colorado Boulder's Cooperative Institute for
Research in Environmental Sciences, and the U.S. Department of Energy (NOAA CIRES 20th Century), the Japanese 55-year
Reanalysis (JRA55), the Modern-Era Retrospective analysis for Research and Applications (MERRA), the Reanalysis I project
from the National Centers for Environmental Prediction and the National Center for Atmospheric Research (NCEP NCAR-
Reanalysis 1), the Reanalysis II project from the NCEP and DOE (NCEP DOE-Reanalysis 2), the NCEP Climate Forecast
System Reanalysis (CFSR), and the WATCH Forcing Data methodology applied to ERA-Interim reanalysis data (WFDEI).
The ERA5 reanalysis, as the latest global reanalysis following ERA-14, ERA-40, and ERA-Interim, provides a comprehensive
field of the global atmosphere, land surface, and ocean waves by assimilating numerous historical observations (e.g., satellite
precipitation data from microwave imagery and few gauge measurements) into the ECMWF Integrated Forecasting System
(IFS) Cy41r2 (Hersbach et al., 2020). The ERA5 reanalysis can simulate the global precipitation with a sophisticated spatial
and temporal resolution with a total of 137 mode layers of 0.01 hPa from 1959 to near real-time. ERA5-land is a re-run of the
land component of ERA5, which is designed to provide a consistent view of land variables over several decades, but with an
enhanced resolution than ERA5 (Muñoz-Sabater et al., 2021). The WFDEI meteorological forcing dataset, however, is
generated based on the ERA-Interim reanalysis after bias correction from gridded observations (i.e., GPCC) and sequential
elevation correction (Weedon et al., 2014). Several classic reanalyses from NCEP are used in our study. NCEP NCAR-
Reanalysis 1 project uses a state-of-the-art forecast system to perform data assimilation during the period 1948-now, while



with a relatively coarse spatial resolution of ~2°, which might cause some errors in small basins upon calculation of basin-
average precipitation (Kistler et al., 2001). We note the precipitation observations are not assimilated into the assimilation
system, so the precipitation from the reanalysis are short-range model forecast accumulations (Janowiak et al., 1998). The
NCEP DOE-Reanalysis 2 is an improved version of the NCEP NCAR-Reanalysis 1, including an updated model with more
realistic physical parameterizations, fixed data assimilation errors, and more digested data (Kanamitsu et al., 2002). The NCEP
DOE-Reanalysis 2 replaces the model precipitation at the land surface with observed data from NCEP/CPC global precipitation
analysis that merges satellite and gauge measurements (Xie and Arkin, 1997). Furthermore, as an important update from NCEP,
the CFSR uses a high-resolution model that is fully coupled with the atmospheric component at a resolution of 38 km with 64
vertical levels from the land surface to 0.26 hPa between 1979 and the present (Saha et al. 2010). Similarly, the CFSR reanalysis
applies the CMAP (Xie and Arkin, 1997) and CPC unified precipitation analysis to reduce the bias derived from the modeled
precipitation in the initial version of NCEP NCAR-Reanalysis 1. Given most analyses only focus on the Earth's status in the
recent half-century, the NOAA CIRES 20th Century project is the first ensemble of sub-daily global atmospheric conditions
spanning over 100 years from 1836 to 2015, providing the best estimate of the weather at any place and time based on the
upgraded data assimilation method, higher resolution, and larger datasets of observations than the previous versions (Slivinski
et al., 2019). We note the NOAA CIRES 20th Century did not incorporate any precipitation observations, meaning the
reanalysis of precipitation is only from the predictions of models. Since the reanalysis provides 80 ensemble members to
constrain the uncertainty fully, we take the ensemble mean as the final precipitation estimate. The JRA55 reanalysis, managed
by Japan Meteorological Agency (JMA), also derives precipitation from remote sensing products combing the model forecasts
since 1958, attempting to provide comprehensive fields of atmosphere to foster the applications in multidecadal variability and
climate change (Kobayashi et al., 2015). The MERRA 2 analysis from the NASA Global Modeling and Assimilation Office
using the GEOS-5.12.4 system covers the period from 1980 to the present with a latency of weeks, with the output resolution
of 0.5°(latitude)×0.625°(longitude). The precipitation from MERRA2 reanalysis follows the assimilation strategy of CFSR,
i.e., consider the CMAP and CPC Unified from NOAA CPC for assimilation. The quality of MERRA2 precipitation has been
evaluated in a previous study, and relatively bad accuracy in high latitudes was reported (Reichle et al., 2017).

We also consider several 'combined products' that merge the above-mentioned data sources, including gauges,

satellites, and reanalysis to estimate precipitation, including the Multi-Source Weighted-Ensemble Precipitation (MSWEP),
Princeton Global Forcings (PGF), and different versions of Global Land Data Assimilation System (GLDAS). The MSWEP
dataset that is featured by full global coverage, high spatial (0.1°) and temporal (3-hourly) resolutions, and distributional bias
corrections optimally merges the precipitation records from gauge measurements (e.g., GPCC), satellite solutions (e.g.,
TRMM), and reanalysis (e.g., JRA55) and achieve better performance than each of the members during the period 1979-now
(Beck et al., 2019). The global and long-term PGF forcing dataset is constructed using the NCEO NCAR-Reanalysis 1 and
multiple observation-based precipitation datasets such as TRMM, GPCP, and CRU TS products to perform the temporal and
spatial downscaling, contributing to the high-resolution precipitation estimations from 1948 to 2016. The GLDAS forcing
dataset generally applies precipitation of different types in different eras. Specifically, GLDAS (v1.0) switches from ECMWF





reanalysis during 1979-1993 to NCEP NCAR-Reanalysis 1 during 1994-1999 and finally uses the CMAP fields from 2001 to
2019 with the NOAA/GDAS atmospheric applied in the year 2000 (Wang et al., 2016). However, the GLDAS (v2.0)
precipitation is from the PGF dataset as the only source from 1948 to 2014. Differently, the GLDAS (v2.1) simulations are
forced with a combination of GDAS, disaggregated daily GPCP precipitation, and AFWA radiation datasets from 2000 to the
present. Please find detailed information about the product version and spatial/temporal resolution in Table 1.
**3.2 Runoff**
Similar to the precipitation, we also collected R datasets from different sources to feed the water balance equation. Firstly, we
collected in-situ discharge measured at the mouths of the rivers from the dataset provided by Dai and Trenberth (2002), namely
the Global River Flow and Continental Discharge Dataset. This observational dataset was compiled from many sources,
including Bodo (2001), NCAR archive, and R-ArcticNET dataset (http://www.R-ArcticNET.sr.unh.edu), and has undergone
the data quality controls during the compilation to avoid errata and inconsistencies. It contains monthly mean volume
observations in 925 major rivers of the world since the 1900s (different rivers have varying lengths) and updates at an irregular
time step (last updated in May 2019). The estimate of global continental freshwater discharge based on the dataset compares
well with alternative estimates and ECMWF reanalysis, though there are some differences among the discharge into the
individual ocean basins. The water volume is converted into the equivalent water depth by dividing the drainage area of the
station. About one-third of the selected 168 river basins are included in this observational dataset, and the missing months
without observation (e.g., after 2019) are set as NA values in the water balance calculation. Apart from this, most of the runoff
datasets used in our study are from a global runoff reconstruction, named Global RUNoff ENSEMBLE (G-RUN ENSEMBLE),
which provides a global runoff reanalysis of monthly runoff rates covering decades to the recent century at a resolution of 0.5°
(Ghiggi et al., 2021). The observation-based G-RUN ENSEMBLE employs the random forest method to learn the runoff
generation using the gridded meteorological observations (precipitation and temperature) with the calibration of the Global
Streamflow Indices and Metadata Archive (GSIM) (Do et al., 2018). The most significant improvement of G-RUN
ENSEMBLE compared to its previous version (GRUN, Ghiggi et al., 2019) is that it considers the forcing uncertainty by
deriving a total of 23 subsets from multiple meteorological reanalysis and observations. Although one of the 23 G-RUN
ENSEMBLE members forced by WATer and global CHange (WATCH) Forcing Data (WFD) only provides the global runoff
data up to December 2001, we still keep it in our study for consistency. It would not influence the water balance estimations
of ET-WB as all the missing months are taken as NA values during calculation. We note an implicit assumption in the
generation of G-RUN ENSEMBLE is that the storage of river water loss can be minimal so the monthly river discharge of the
river mouth equals the average catchment runoff depth. Given that the G-RUN ENSEMBLE is only calibrated from small
catchments with areas ranging from 10 to 2,500 km$^2$, this assumption might not be strictly valid for large river basins, although
it has shown comparable performance with several global runoff simulations and reconstructions like the Global Drought and
Flood Catalog (GDFC) (He et al., 2020) and ERA5. Moreover, the human activities, including human water use and reservoir
management, lack a physical-based representation in the random forest machine learning method (but implicitly considered



during the model training), and the apparent outliers caused by human activities (e.g., an abrupt decrease of river discharge
after dam construction) have been removed. Therefore, we additionally compare the R datasets used in our study (mainly from
G-RUN ENSEMBLE) with the streamflow records from the GRDC archive in 53 river basins worldwide since they are the
only regions where the discharge observations are available with the spatial and temporal consistency of our study (Table S3).
A satisfactory performance of the estimations in the levels of multi-mean and long-term trends is found, which are the focus
of our study and the relevant future applications (Fig. S1). We also used a synthesized global gridded runoff product that
merges runoff estimates from different global hydrological models (GHM) constrained by hydrological observations using an
optimal weighting method during 1980-2012 (namely Linear Optimal Runoff Aggregate, LORA), which works dynamically
based on the comparisons with in-situ data when accounting for the variance among members (Hobeichi et al., 2019). The
LORA product, with a consistent spatial resolution of 0.5°, is also used as the benchmarking dataset for G-RUN ENSEMBLE
and achieved similar performance. A similar limitation is shared in these global gridded runoff reconstructions, i.e., the
neglection of river routing, which may lead to an overestimation in the computed uncertainties over large basins. In addition,
since the LORA is the merged result from eight GHMs with different physical structures and model parameterization schemes,
the representation of the basins with significant anthropogenic activities should be taken with caution. For example, there is a
low observed runoff of ~0 across the regions having high irrigation areas and/or artificial surfaces. As an important member
of the LORA dataset, the WaterGAP Global Hydrology Model (WGHM), providing the global water resources dynamics from
1901-2016 at a 0.5° resolution (Müller Schmied et al., 2021), is also selected in our study for the computation of ET-WB. The
most recent version (2.2d) of the WaterGAP framework consists of five water use models, including irrigation, livestock,
domestic, manufacturing, and thermal power sections, the linking model that computes net abstractions from groundwater and
surface water, and the WaterGAP Global Hydrology Model (Müller Schmied et al., 2021). The discharge simulations are
applied in the water balance calculation, which was forced by WFDEI precipitation during the study period and considered the
human effects such as dam management. The river routing schemes follow Döll et al. (2014), where water is routed through
the storages depending on the fraction of surface water bodies. The state-of-the-art global river discharge reanalysis, the Global
Flood Awareness System (GloFAS), serves as a significant supplement to the R inputs in water balance. The GloFAS system
simulates the global discharge by coupling runoff simulations from the specific model forced with the ERA5 reanalysis and a
channel routing model. The GloFAS product aims to provide daily high-resolution (0.1°) gridded river discharge forecasts
from 1979 to near real-time. Different versions of GloFAS reanalysis are used in our study, where the main differences are
from the hydrological modeling scheme. For example, the GloFAS (version-2.1) applies a combination of the Hydrology Tiled
ECMWF Scheme for Surface Exchanges over Land (HTESSEL) land surface model with the LISFLOOD hydrological and
channel routing model (Harrigan et al., 2019). The surface and subsurface runoff from the HTESSEL are used as input for the
LISFLOOD model (Hirpa et al., 2018). For the newer versions like 3.0 and 3.1, both the runoff generation and routing
processes are based on the full configuration of the LISFLOOD model, the former of which is an offline version provided by
Alfieri et al. (2020), and the latter is an operational online version that was released in early 2020 with some changes in web
and data services. Despite this, we take both into consideration as they are the only datasets providing near-real-time discharge



information. All the versions of GloFAS used in our study have been calibrated by more than 1,200 gauge stations worldwide,
which greatly improves the performance than those without any calibrations (Alfieri et al., 2020). Some procedures are needed
for discharge-type R datasets (i.e., WGHM and GloFAS-family products) to find the grid cell coinciding with the river mouth
of the basin. For example, we find the certain grid with the maximum drainage area within the basin based on the static total
upstream area file provided by GloFAS, which is defined as the catchment area for each river segment (i.e., the total area that
contributes to water to the river at the specific grid point). Then, the discharge forecast of that grid point should be divided by
the corresponding drainage area to be converted into equivalent water depth. For the global land, the total freshwater flowing
into the ocean is estimated as the sum of the discharge of all the coastal grid cells based on a mask at the corresponding
resolution (e.g., 0.1° for GloFAS). As such, the differences in the spatial resolution (e.g., 0.5° for the WGHM and 0.1° for the
GloFAS) can contribute to some discrepancies in the final estimates of R. Finally, it is worth mentioning that we manually set
the R-value as zero for the 13 endorheic basins without runoff flowing into the ocean, except for Volga, Ural, and Kura River
basins that flow into the Caspian Sea (Fig. 1 and Table S2).

**3.3 Terrestrial water storage**

Seven global terrestrial water storage datasets are used to derive ΔS and input the water balance equation. Six of these TWS
datasets are GRACE solutions and one is from the WGHM. The GRACE mission has been the preferable tool to assess the
large-scale variations in terrestrial water storage at a near-monthly scale from 2002 to 2017, with the GRACE Follow-On
successor satellite launched in 2018 (Tapley et al., 2004; Kornfeld et al., 2019). There are generally two classes of methods to
retrieve TWS anomaly signals from GRACE measurements, the spherical harmonic (SH) and the mass concentration blocks
(mascon) methods. The SH method is a standard for the first decade of the GRACE era, which is processed by parameterizing
the global time-varying gravity field using SH coefficients (Wahr et al., 1998). However, such a method should undergo a
series of post-processing of the truncation of degree/order in SH coefficients, spatial smoothing, de-correlation filtering, and
replacement of low-degree coefficients, etc. Various background models, such as glacial isostatic adjustment and de-aliasing,
should also be considered. Therefore, different methods have been developed to restore the signal leakage and bias introduced
during the post-processing. These methods include additive and multiplicative approaches, model-based scaling factors, data-
driven methods, and constrained and unconstrained forward modeling methods (Long et al., 2015; Chen et al., 2019;
Vishwakarma et al., 2017). However, the mascon method has provided another user-friendly option for the community in
recent years, which functions by parameterizing the Earth's gravity field with the regional mass concentration functions. This
kind of method does not need substantial post-processing techniques for signal restoration and can attenuate the noise during
the gravity inversion process through regularization of the solution (Save et al., 2016; Xiong et al., 2022a). So the increasing
attention in the non-geophysical community has been attracted by the mascon solution over the years (Abhishek et al., 2021).
However, it is noticed that different GRACE ground system institutions can perform the post-processing for the fundamental
level-1 GRACE data using different strategies, for example, the varying algorithms to the effect of glacial isostatic adjustment
and the regularization or stabilization of the regional mass concentration functions may affect the hydrological analysis at



smaller scales (<~3°) (Scanlon et al., 2018; Watkins et al., 2015; Vishwakarma, 2020). In this case, we collected the latest
Release Version 06 level-2 SH solutions from different official GRACE processing agencies, including the University of Texas
Center for Space Research (CSR), NASA's Jet Propulsion Laboratory (JPL), and GeoforschungsZentrum Potsdam (GFZ), as
well as three level-3 mascon solutions from CSR, JPL, and NASA's Goddard Space Flight Center (GSFC) during the period
April 2002-December 2021, which is the longest time span that GRACE (and GRACE Follow-On) can achieve at the present
stage. The signal leakage and bias in three SH solutions are corrected using the forward modeling method, with the above-
mentioned standard processing procedures performed (Swenson and Wahr, 2006). The mascon JPL solution that employs a
Coastal Resolution Improvement (CRI) filter that reduces signal leakage errors across coastlines has undergone the adjustment
from official scaling factors based on the CLM land surface model (LSM) (Wiese et al., 2016). As previously mentioned at
the beginning of the Data section, the inconsistent spatial resolution of different mascon solutions will not impact the ET-WB
calculations as we only perform the water balance budget at the basin (and global) scale (Save et al., 2016; Loomis et al., 2019).
The 33 missing months due to the data gap between two generations of GRACE missions and instrumental issues have been
statistically interpolated using a recently proposed method based on the Singular Spectrum Analysis method (Yi and Sneeuw,
2021). This method can infer missing data from long-term and oscillatory changes extracted from available observations and
does not rely on any external forcing, thus avoiding the uncertainty introduced by other datasets (e.g., precipitation).

Apart from the GRACE solutions, the simulations from the WGHM model are also used to avoid the strong correlation
among GRACE solutions and provide a potential alternative viewpoint. The WGHM simulations of TWS include most of the
key components in the land system, including canopy, snow and ice, soil moisture, groundwater, and surface water bodies
(e.g., river, lake, wetlands, and reservoirs). However, the glacier water storage is not simulated in WGHM, which might induce
some errors in high-latitude cold regions (Müller Schmied et al., 2021). The major human interventions such as dam
management and human water use are also considered, which have been reported to greatly impact the regional terrestrial
water storage balance (Rodell et al., 2009). This is the main advantage of the selected WGHM over other widely used
GHMs/LSMs, such as GLDAS VIC and Noah models. GRACE solutions generally provide the anomalies of TWS relative to
a long-term mean, but the WGHM simulates the actual value of TWS. However, this will not affect our derivation for the ΔS
and the subsequent ET-WB estimations.
**3.4 Evaporation**
Benchmarking ET-WB against other global ET products is crucial to evaluate its performance. With the principle of 'different
types of datasets have their unique values' in mind, four different categories of auxiliary ET products have been chosen for
comparison with ET-WB at multiple time and space scales. These include the MODIS Global Evapotranspiration Project
(MOD16A2), the FLUXCOM ensemble dataset, the Global Land Evaporation Amsterdam Model (GLEAM), and the
simulations from WGHM. The MOD16A2 product estimates the terrestrial ET as the sum of evaporation from soil and canopy
layer and the transpiration from plant leaves and stems (Mu et al., 2011). This satellite-based dataset is estimated under the
framework of the Penman-Monteith equation with the effective surface resistance to the evaporation from the land surface and



transpiration from plant canopy, which is estimated based on the MODIS remotely sensed data including surface albedo, land
cover classification, and vegetation information. The MOD16A2 dataset was originally produced at a spatial resolution of 1km
and a temporal resolution of 8-day from 2000-2014. However, we used the re-processed monthly 0.5° product provided by the
Numerical Terradynamic Simulation Group (NTSG) at the University of Montana
(http://files.ntsg.umt.edu/data/NTSG_Products/MOD16/). The FLUXCOM "remote-sensing" database ("RS" setup) employs
nine machine learning algorithms to integrate ~20,000 flux observations across the globe with the satellite-based predictors
from the MODIS mission (Jung et al., 2019). Therefore, it is considered an observation-driven product of three energy balance
variables, namely, net radiation, latent energy, and sensible heat. Nonetheless, the product is subject to uncertainty in the choice
of prediction models and is also limited in spatial/temporal resolution (0.0833°/8-daily) and time coverage (2001-2015) of the
satellite inputs. Similarly, we used the re-processed monthly version of the product with a resolution of 0.5° by spatial and
temporal aggregation, which is the median value of the ensemble members per grid cell and month. A key difference between
the FLUXCOM and other ET datasets is that the former focuses only on the vegetated region because of the lack of eddy tower
observations in these regions, meaning the ET values in unvegetated (barren, permanent snow or ice, water) area was omitted.
We convert the latent energy data to ET by dividing it with the latent heat of vaporization, a constant value of 2.45 MJ/kg (or
multiplying 0.408 kg/MJ) or 28.35 W/m$^2$. We note the FLUXCOM database also develops the "RS+METEO" setup that uses
daily meteorological data and mean seasonal cycles of satellite data with three machine-learning approaches. Since the
differences between these two setups over global basins are still unclear, and beyond the scope of our study, only the "RS"
setup is chosen for comparison and demonstration with ET-WB. It needs to be mentioned that we did not use the in-situ
measurements from the regional FLUXNET eddy covariance towers because of the uneven and sparse distribution from a
global perspective, which is not consistent with the spatial scale of ET-WB. In addition, the GLEAM model estimates the
terrestrial ET separately, which comprises the individual components of transpiration, interception loss, bare soil evaporation,
snow sublimation, and open-water evaporation (Martens et al., 2017). It firstly estimates the potential ET using the Priestley-
Taylor equation based on satellite observations of surface net radiation and near-surface air temperature, then converts the
potential ET to actual ET using the evaporative stress factor, which is estimated from the remote sensing vegetation microwave
vegetation optical depth and predicted root-zone soil moisture from a water balance model. The GLEAM is more inclined to
a 'reanalysis' dataset as it does not use the satellite observations directly (like MOD16A2) but indirectly includes the satellite
observations to estimate ET. Similar to the FLUXCOM dataset, the GLEAM product also has two sub-versions, 'a' and 'b',
with the main difference in the time span (1981-2021 for 'a' and 2003-2021 for 'b') due to different inputs considered. We
choose version 3.6a to compare with ET-WB. Finally, the hydrological simulations of ET from WGHM are also included for
data consistency, which was previously used to contribute to the runoff, terrestrial water storage, and precipitation (WFDEI
forcing) estimations. Moreover, an alternative source (GHM) of ET can also strengthen the justification upon the comparison
with derived ET-WB.



**4 Results**
**4.1 Global evaluation of ET-WB**
**4.1.1 Monthly assessment**
Comparison and analyses of ET-WB and auxiliary ET datasets are carried out at various temporal scales to examine the
reliability of ET-WB comprehensively. The long-term average seasonal cycle of ET during the period 2002-2021 is detected
over global land (Fig. 3a). A clear unimodal distribution is observed with the highest ET in July (median value: 65.61
mm/month (mm/m)) and the lowest result in January (median value: 36.11 mm/m) based on ET-WB, with the spread range of
roughly ±10 mm/m from different subsets of the ensemble. Furthermore, the seasonal cycle of other ET products is generally
within the range of ET-WB ensemble with similar intra-annual characteristics. All of the GLEAM, MODIS, and WGHM data
illustrate an overestimation of ET from March to June and an underestimation between September and November compared
with the median values of ET-WB, but they are completely within the range of the ET-WB ensemble. Nevertheless, the
FLUXCOM product tends to have higher ET than ET-WB due to the fact that FLUXCOM only considers the ET in the
vegetated regions, and the unvegetated areas, such as those in the deserts of Sahara and Qinghai-Tibetan Plateau are masked
(Jung et al., 2019). This would subsequently influence our comparisons in basins with a certain proportion of unvegetated area
and the global land.
The seasonal pattern of ET-WB is highly consistent with that of precipitation in both amplitude and periodicity, which
generally increase from the beginning to the middle of a year, followed by a gradual decrease. This contemporaneous relation
between ET and P without time lag is also revealed by Rodell et al. (2015). However, the spread range in P is wider than ET-
WB, meaning it is an important contributor to the uncertainty of the ET-WB, especially in water-limited months like February,
April, and November (Fig. 3b). In addition, we also found that the seasonal cycle of ΔS presents a reverse distribution than
other water components (e.g., P and R), in which ΔS decreases from positive to negative in the first half of the year (January
to June) and then slowly rebound until the end of the year. In other words, the land system is losing water from April to October
and gaining water until April of next year, implying a significant time lag between terrestrial water storage and P on a global
scale (Fig. 3c). The narrow spread range of ΔS is attributed to the high agreement between the six GRACE solutions used, not
showing the real uncertainty of TWSA (ΔS) estimates. Counterintuitively, P lags R by two months, possibly related to the
snowpack immobilization and the strength of summer convective rainfall in high-latitude regions (Rodell et al., 2015).
Additionally, R demonstrates an interesting distribution with a constrained change range in all months with a few
overestimations. It should be stemming from the reduced uncertainty in the choice of R datasets because we used the 23 (out
of 29) G-RUN ENSEMBLE subsets that were generated using the same model but forced by different forcing, together with
the interventions from other datasets (e.g., GloFAS reanalysis) (Fig. 3d).
Multiple statistical metrics are used to quantify the relative performance of the ET-WB product, which are calculated
using the ensemble median ET and other global ET datasets. Global examinations of the relative bias (RB) based on different
auxiliary datasets on the monthly scale indicate an overall agreement with ET-WB, with most (74%, 63%, 57%, and 77% for





GLEAM, FLUXCOM, MODIS, and WGHM, respectively) river basins having RB between -20% and 20% (Fig. 4). For the
global land, the RB reaches 1.22%, -17.31%, -3.68%, and 2.96% for above four products, correspondingly, but with strong
spatial heterogeneity among basins. Specifically, widespread overestimation of ET-WB than other datasets are reported in East
Europe, West Russia, South and East Asia, and West Australia, with the maximum RB of nearly 300% in the Ashburton River
basin (ID: 138) of Australia based on the MODIS ET dataset. On the contrary, the consistent underestimation of ET-WB
compared with other products is also seen in West Europe, East Russia, and Southeastern basins of Australia, where RB is
mostly small. However, divergent patterns of different ET datasets in parts of South and North America, Africa, and Central
Asia highlight inherent uncertainty in each product and that it is impossible to have a single best-performing ET dataset for
the whole globe. However, the RB values of ET-WB are within the range of ±20%, meaning the ET-WB is comparable to
these ET products and, therefore, can serve as an independent benchmarking product (Figs. 4a, 4c, 4e, and 4g). Alternative
metrics like CC and NSE provide additional insights. Relatively better performance of ET-WB is apparent in the humid basins
of high-latitude Eurasia, North America, and South China according to the comparably higher CC (>0.8) and NSE (>0.4) than
other regions like South America and Africa (Figs. S2 and S3). This might be due to better simulation accuracy of, for example,
reanalysis and GHMs, in humid zones than in arid regions. Though the reported NSE value may not appeal as satisfactory in
an absolute sense, it only represents the median ET-WB. Distinctive choice of ET subset over different regions may lead to
improved results, albeit without informing the full spread of the uncertainties. Additionally, RMSE results further convey
higher errors of ET-WB in smaller regions than in larger ones (Fig. S4) because of the reduced retrieval errors of GRACE
solutions as the basin size increases (Scanlon et al., 2018). The notable exception is the Amazon River basin (ID:1), which
shows inconsistency between ET-WB and different ancillary products (e.g., GLEAM and MODIS). It is similar to a recent
regional study (Baker et al., 2021), although a strong agreement between water balance ET and shortwave radiation was
observed. For all the 168 basins, the scatter plots illustrate a reasonable agreement between ET-WB and multiple ET datasets
(Figs. 4b, 4d, 4f, 4h). Despite the very small RB (from 0.09% of WGHM to -7.96% of MODIS), the skewed estimates are
discovered in high-ET periods and regions, while most points having small ET values are perfectly located around the 1:1 line.
Another discrepancy between ET-WB and other datasets is the existence of negative values of the former primarily in high-
ET regions/periods, which is very likely resulting from the non-closure error among various water balance datasets (Pan et al.,
2017; Rodell et al., 2011; Lehman et al., 2022) along with their respective shortcomings (e.g., non-consideration of river
routing in G-RUN Ensemble runoff data) and should be delved into in future studies.




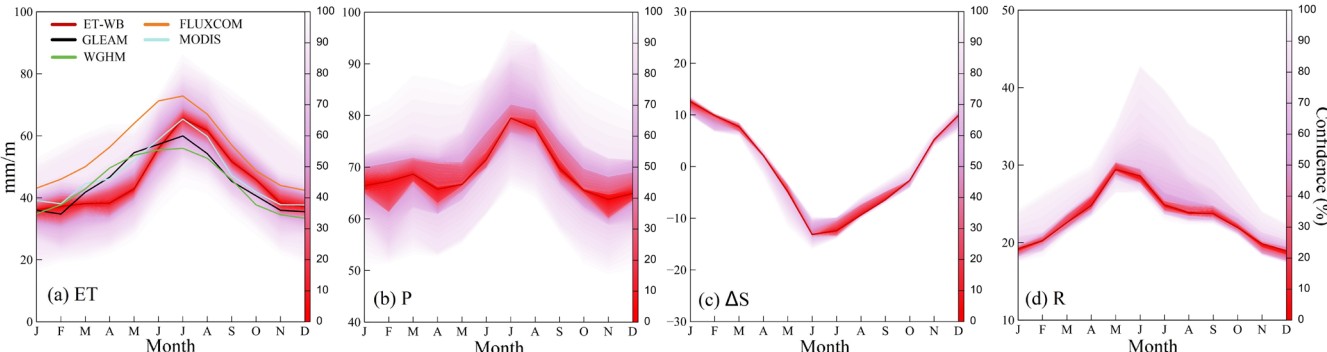


**Figure 3: Monthly average values of the ET-WB and multiple auxiliary ET products as well as other water components over global land during the period 2002-2021. The shading shows the spread range among different datasets.**





**Figure 4: Comparisons between the ET-WB and multiple auxiliary ET products (a, b: GLEAM; c, d: FLUXCOM; e, f: MODIS; g, h: WGHM) on a monthly scale during the period 2002-2021. The left column represents the global distribution of RB, and the right column represents the corresponding scatter plots. The color of the scatter points indicates the kernel density.**

**4.1.2 Annual assessment**

Inter-annual variability of ET and related water balance components are also examined over global land (Fig. 5). There are generally three episodes shown in the ET-WB dataset. These include a gradual increase from 2003 to 2010 and a subsequent decrease during 2010-2015, followed by a sharper reduction in the remaining years (Fig. 5a). A large inter-ensemble range, which aggravates during the recent time periods, due to the propagation of errors in monthly estimations of water balance ET is found. Other ET datasets, despite the different time spans, still present a similar variability to ET-WB with the overestimations in MODIS and FLUXCOM. As discussed above, the significant differences from FLUXCOM can be attributed to the specific data generation method. Furthermore, the annual variations of ET are typically explained by the changes in P, which experienced an increasing trend during 2003-2010, followed by an abrupt decrease between 2010 and 2015 (Fig. 5b). However, the increase of P during 2015-2021 does not directly translate to the enhancement of ET based on ET-WB results, though the GLEAM shows a more 'reasonable' increase under the assumption of the limited influence of the human interventions on the global ET on an annual scale. This inconsistent phenomenon is because of the significant increase of R values since 2015 (particularly in 2020 and 2021), which are mainly driven by GloFAS reanalysis data as the 23 G-RUN ENSEMBLE subsets are not available from 2020 (Fig. 5d). Therefore, the overestimation of R in GloFAS data can explain the abrupt change in ET-WB over recent years, implying that caution should be taken when interpolating the ET-WB results after 2019 due to the availability of the limited dataset. This is not only because of the controlling role of specific water components in ET-WB (e.g., a wide range of P similar to ET) but also the limited data availability due to delayed updates (e.g., G-RUN ENSEMBLE). Moreover, ΔS does not play a crucial role on an annual scale because of the relatively small amplitude and the confident estimations of GRACE signals in such a large area (Fig. 5c).

Statistical metrics are re-assessed on an annual scale to evaluate the differing performance of ET-WB across temporal scales. A similar spatial pattern is revealed according to the RB results but slightly degrades over most basins, which is seemingly caused by error accumulation from water components and the relatively short time span for calculation (e.g., 19 years) (Fig. 6). For the global land, the RB reaches -0.05%, -18.07%, -4.61%, and 1.73% for the GLEAM, FLUXCOM, MODIS, and WGHM, respectively. Alternate metrics such as CC and NSE also indicate deteriorating accuracy of ET-WB after converting from monthly to the annual time scale for the single basin, while RMSE is improved if we use the same unit (Figs. S5-S7). However, the scatter plots of annual ET in a total of 168 basins between ET-WB and auxiliary datasets show significant improvements to that on the monthly scale due to the offsets of negative ET values within a year and more benign fluctuations of annual ET than the monthly series. For example, the fitted slope of the regression between ET-WB and other datasets is 0.92 (GLEAM), 1.03 (FLUXCOM), 0.93 (MODIS), and 1.01 (WGHM), respectively, with higher CC and NSE compared with their monthly counterparts.




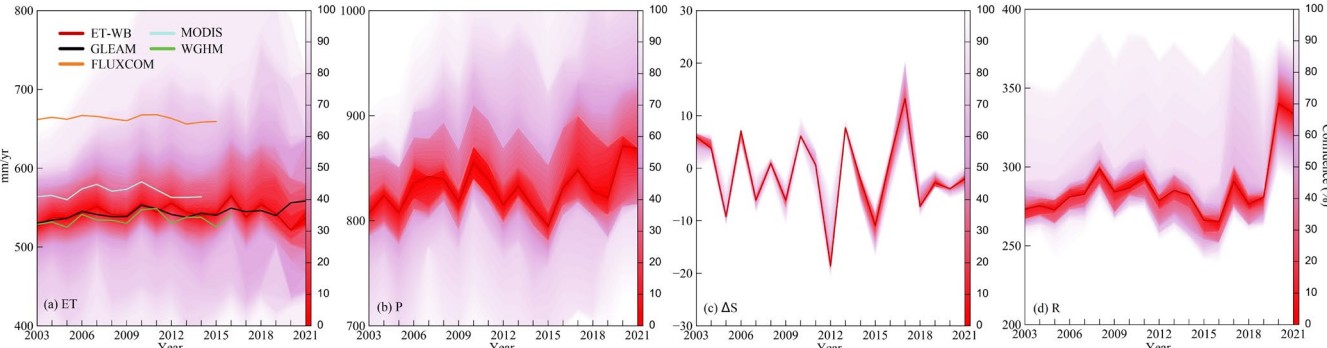


**Figure 5: Annual time series of ET-WB and multiple auxiliary ET products as well as other water components over global land**
**during the period 2003-2021. The ET in 2002 is excluded from the calculation because of the missing values from January to April**
**2002. The shading shows the spread range among different datasets.**










**Figure 6: Comparisons between the ET-WB and multiple auxiliary ET products (a, b: GLEAM; c, d: FLUXCOM; e, f: MODIS; g, h: WGHM) on the annual scale during the period 2002-2021. The left column represents the global distribution of RB, and the right column represents the corresponding scatter plots. The color of the scatter points indicates the kernel density.**

## 4.2 Spatiotemporal variation of ET-WB

Spatiotemporal variability of ET from the ET-WB and other auxiliary ET products are assessed for comparison. The long-term mean of annual ET based on the ET-WB illustrates a clear spatial pattern, with relatively higher ET in humid zones of South America, Eastern North America, central South Africa, and South Asia, while the lower ET in arid regions of Western United States, North and South of Africa, Central Asia, and Australia (Fig. 7a). Specifically, the Kapuas River basin (ID: 131) in Indonesia has the highest ET-WB flux of 1565 mm/yr due to the hot and humid climate regionally (Hidayat et al., 2017). The endorheic Tarim River basin (ID: 14) in northwest China has the lowest annual ET of 127 mm/yr among 168 study basins because of the prevailing extremely dry climatic conditions. The homogeneous spatial patterns between ET-WB and GLEAM, FLUXCOM, and MODIS products can further validate the reliability of ET-WB (Fig. S8). In addition, WGHM reports a slightly different distribution from the other three datasets and ET-WB, which can result from modeling uncertainty due to simplified model parameterization and the un-calibrated ET simulations (Müller Schmied et al., 2021). Specifically, we observe the consistent overestimations of ET-WB than other datasets in East Europe, West Russia, South and East Asia, and West Australia, especially in the wet areas like the Yangtze (ID: 13) and Mekong (ID: 31) River basins. On the contrary, relative underestimations are observed in West Europe, East Russia, and Southeastern basins of Australia (Fig. 7). The divergent patterns between ET-WB and different datasets are seen in large-scale regions of South and North America, Africa, and Central Asia. Nevertheless, the regional differences are mostly within the range of ±100 mm/yr, which is a relatively small range for basins with higher ET values, unlike the dry basins with relatively small ET (Figs. 7c-7f). The spatial distributions of differences between ET-WB and other datasets are similar to the RB results (Fig. 4), which manifests from the homologous calculation formula (Eq. 5). For the global land, the long-term mean annual ET estimates from ET-WB are concentrated within the range of 500-600 mm/yr among ensemble members, with the median estimates of 549 mm/yr (Fig. 7b). This number is comparable to the result from GLEAM (543 mm/yr), MODIS (569 mm/yr), and WGHM (534 mm/yr). The relatively higher value of global ET from FLUXCOM (663 mm/yr) is attributable to the exclusion of the unvegetated area in the global averaging, while it has shown good agreement with several global products (e.g., GLEAM) in the vegetated area (Jung et al., 2019).

The annual trends of ET from various datasets during 2003-2014 are assessed. The calculation period is selected to be consistent with the temporal span of different products, which can cause some biases in determining trends due to the relatively short computation period (i.e., 12 years). The ensemble median results of the ET-WB ensemble reveal a spatial distribution with the increasing ET detected in South America (around the Amazon River basin), Europe, East Russia, South and East Asia, South and North Africa, and Australia. Over these regions, the Burdekin River basin (ID: 94) in Australia has the most rapid growth rate of 31.4 mm/yr$^2$, which is about 100 times the slowest increasing slope (0.3 mm/yr$^2$) in the Alazeya





River basin (ID: 165) of Russia (Fig. 8a). Significant depletion of ET is observed in the central North America and Africa
continents as well as West Russia with the lowest trend of -22.8 mm/yr$^2$ in the Moose River basin (ID: 107) of Canada. We
also noticed similar spatial patterns based on other auxiliary ET datasets (Fig. S9), however, with the differences in the
magnitudes of trends. Such differences are reasonable because the trend estimations contain uncertainty in a short 12-year long
period, let alone the errors inherent to various products. Therefore, we see an interesting spatial distribution of the differences
between ET-WB and other datasets (Figs. 8c-8f), where the regional differences in trends are similar to the actual trend
summarized by the corresponding dataset (Fig. S9). In particular, ET-WB is prone to overestimate the trends for regions with
increasing ET, and the overestimations are larger if the trends are larger (based on other ET datasets), and vice versa. In a
nutshell, unlike TWS/P-based evaluation (Held and Soden 2006; Xiong et al., 2022b), the 'dry gets drier and wet gets wetter'
paradigm can be typically inferred from ET-WB on a basin scale, which generally exaggerates the prevailing
increasing/decreasing ET tendencies in the basins (Yang et al., 2019). On a global scale, the median value of trend estimates
from ET-WB ensemble members is 1 mm/yr$^2$, very close to the results from GLEAM (0.8 mm/yr$^2$) and WGHM (0.8 mm/yr$^2$).
However, both FLUXCOM and MODIS report small negative values of -0.3 and -0.1 mm/yr$^2$, respectively, which still fall
within the spread range of the ET-WB ensemble estimations (Fig. 8b).




**Figure 7: Global distribution of (a) the long-term mean in annual ET-WB and (c-f) its difference with multiple auxiliary ET products during 2003-2021. The long-term mean is calculated as the sum of the long-term averages of ET in each month. Subplot (b) shows the histogram and the probability density distribution of the ET-WB ensemble results over global land.**




**Figure 8: Same as Figure 7, but for the annual trends. The ET in 2002 and after 2014 are excluded from the calculation because of the missing values of GRACE data in 2002 and the missing values of MODIS product after 2014. The trend is calculated by using Sen's slope method. Subplot (b) shows the histogram and the probability density distribution of the ET-WB ensemble results over global land.**

## 4.3 Uncertainty in ET-WB

Quantification and attribution of uncertainty in the ET-WB ensemble play important roles in the justification and potential usages of the proposed dataset. Based on the methods described in Section 2.3, we present the global distribution of the RMS values of uncertainty in ET-WB and related water components as well as the auxiliary ET products (Fig. 9). We observe a clear spatial pattern of the uncertainty, which generally increases along with the reduction in basin size. Several large-size basins, such as Ob (ID: 5), Yenisey (ID: 7), and Lena (ID: 9) River basins, possess a lower uncertainty (<20 mm/m) compared to those medium-size basins like Mekong (ID: 31) and Ganges (ID: 22) River basins where uncertainties in ET-WB are between



40 and 80 mm/m. However, the small-size basins suffer from substantial uncertainties in ET-WB, even exceeding 100 mm/m
in some regions of mainland Australia and Europe (Fig. 9). The worst phenomenon happens in the Essequibo River basin (ID:
156), with the RMS of the uncertainty of 267 mm/m primarily arising from the high uncertainties in GRACE data (Fig. 9a). A
seemingly more optimistic situation is observed from the uncertainty of four auxiliary ET products, where the low-latitude
humid zones apparently suffer from higher uncertainty than the high-latitude regions, though they are essentially smaller than
30 mm/m with the maximum of 65 mm/m in the Ogooue River basin (ID: 68) of Gabon (Fig. 9c). It is not surprising because
the uncertainty in ET-WB is propagated from three water components including P, $\Delta$S, and R, but that in the auxiliary ET
products in our study is calculated as the standard deviation among four datasets. Despite this, the performance of ET-WB
over large basins is still comparable to these ET datasets, whose uncertainties share similar spatial distribution with P to a
certain degree. As an important input for GHM and some other ET products (e.g., "RS+METEO" setup of FLUXCOM), P can
determine the actual performance of the auxiliary ET products. It can even determine the uncertainty in R datasets which
subsequently contributes to the uncertainty of G-RUN ENSEMBLE; the main data for our water balance forcing (Figs. 9d and
9f). However, the "reduction-with-increasing-size" pattern of uncertainty in ET-WB seems more relevant to the uncertainty in
$\Delta$S datasets, which is from six different GRACE solutions and a set of simulations from WGHM. It has been widely reported
that the retrieval bias of GRACE missions is higher in smaller regions due to the coarse spatial resolution and the pronounced
signal leakage effects (Scanlon et al., 2018) (Fig. 9e). This is contended to be the main reason for the similar distribution and
amplitudes of uncertainty in $\Delta$S and ET-WB for smaller basins, while the uncertainty in ET-WB over larger basins is mainly
controlled by other factors like P. However, over a global scale, the uncertainty of ET-WB that roughly fluctuates below 15
mm/m (RMS: 9.7 mm/m) is controlled by that of P (RMS: 8.3 mm/m), the uncertainty in $\Delta$S is relatively small because of the
very large area (Fig. 9b). The sharp increase in uncertainty of R from the year 2020 is caused by the unavailability of 23 G-
RUN ENSEMBLE datasets. Similarly, the abrupt decrease of uncertainty in auxiliary ET products after 2015 is due to the
limited time coverage of FLUXCOM and MODIS products, with an RMS of 5.3 mm/m over the whole period. This different
behavior underscores the potential users to pay attention to the number of datasets used to produce ET-WB. In addition, ET-
WB will be updated as the new/updated versions of these constituent datasets are released to constrain such uncertainties.

To further investigate the influential factors to the uncertainty in multiple variables, the relationship between the

uncertainty and basin size, climate conditions (represented as the long-term mean AI), and human interventions (represented
as the irrigation rate, which is defined as the equipped irrigation area versus the basin area) are detected (Fig. 10). As we
described above, the obvious relationship between uncertainty in $\Delta$S and basin size governs the increasing uncertainty of ET-
WB along with the enhancement of basin area, while the uncertainty in auxiliary ET products generally keep at a lower level
of uncertainty similar to P and R (Fig. 10a). Although other variables like P and R do not show any pattern associated to the
basin area, they present favorable dependence upon the aridity of the basin, where they are inclined to have higher uncertainty
in more humid regions with higher AI (Fig. 10b). No clear pattern between ET uncertainty and irrigation area can be apparently
deduced, whereas it is worth mentioning that the significant irrigation equipped for groundwater resources can lead to
significant short-term and long-term variations of, for example, $\Delta$S and R, which is the case in some basins in North China
(e.g., Haihe River basin, ID: 67) and North India (e.g., Indus River basin, ID: 27) (Fig. 10c). The human-induced inordinate
fluctuations of water balance (e.g., through reservoir management, groundwater extraction) can influence the quality of ET-
WB by impacting the accuracy of the specific forcing variable (e.g., R). Finally, the uncertainty in ET-WB can be further
intensified for the small wet basins with significant human disturbance, so caution should be particularly taken when drawing
scientific conclusions using ET-WB in those regions.

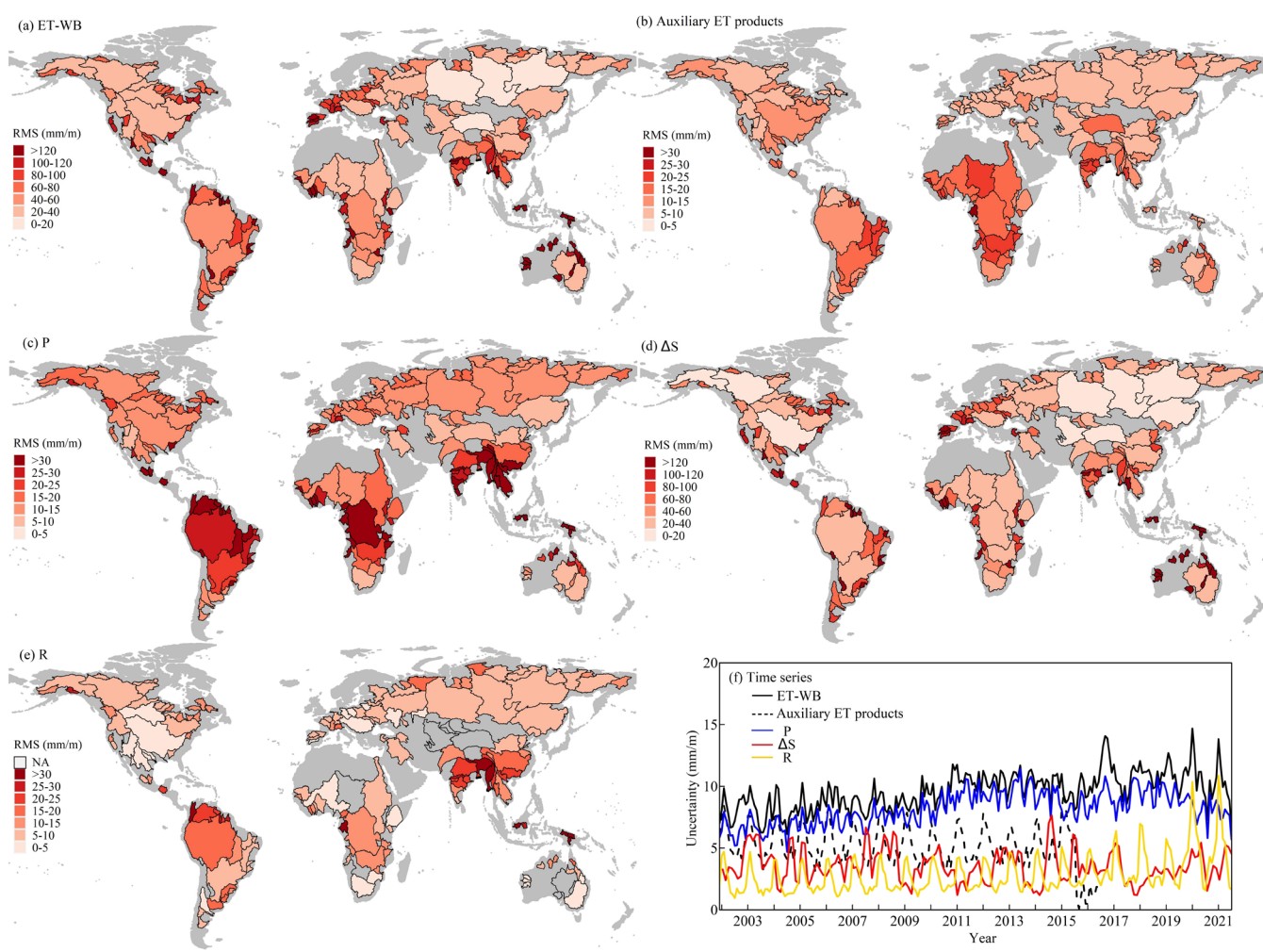


**Figure 9: RMS of uncertainty in the ET-WB and different water components over global basins. Subplot (b) shows the time series**
**of uncertainty in different variables over global land. The NA values in sub-plot (f) R are because the runoff is manually set as zero**
**in these regions. Please refer to the Data section for details.**



**Figure 10: Relationship between RMS of uncertainty in the ET-WB, auxiliary ET products, different water balance components, and (a) size, (b) aridity index, and (c) irrigation rate of the basins.**

## 5 Discussions

### 5.1 Comparisons with previous regional studies

Although a global compilation of water balance estimations of ET is still lacking, previous regional studies have demonstrated the applicability of the water balance ET at different basins of the world. Comparisons with such regional studies are beneficial to the benchmark of ET-WB. Rodell et al. (2004) initially proposed the plan to retrieve ET on basin scales based on the water



balance model and early GRACE data and applied it in the Mississippi River basin (ID: 4) from July 2002-November 2003.
By comparing with model predictions of ET, the RMS differences between water balance ET and GLDAS, GRDS, and
ECMWF-based ET were found to be 0.83, 0.67, and 0.65 mm/day (equivalent to 24.9, 20.1, and 19.5 mm/m), respectively
(Rodell et al., 2004), which are comparable to our RMSE results on the monthly scale, i.e., 19.46 (GLEAM), 18.41
(FLUXCOM), 24.29 (MODIS), and 23.04 (WGHM) mm/m. Given the significance of the water balance method in ungauged
regions, several studies have tested its performance in the data-sparse Tibetan Plateau (Xue et al., 2013; Li et al., 2014; Li et
al., 2019). For example, Xue et al. (2013) compared four ET products, including GLDSA, JRA, MODIS, and Zhang_ET
(Zhang et al., 2010), against the water balance ET in the upper Yellow (ID: 24) and Yangtze (ID: 13) River basins, revealing
the overestimations of GLEAM and MODIS relative to the water balance ET. These comparisons are similar to the RB
examinations in our study based on ET-WB. As the largest river basin of India that accounts for 26% of the country's landmass,
the Ganges River basin (ID: 22) shows a mean monthly average ET of 63.2 mm/m (Syed et al., 2014), which is comparable to
60.9 mm/m calculated in our study despite the different study periods. A case study in the Volta River basin (ID: 46) of Africa
reported the annual fluctuations of water balance ET ranging from 700 to 800 mm/yr during the period 2004-2011 (Andam-
Akorful et al., 2015), relatively lower than the long-term mean ET-WB of 830 mm/yr. The relative accuracy of water balance
ET in the exorheic river basins of China has also been previously evaluated. For example, Zhong et al. (2020) employed the
water balance equation to estimate regional ET and compared them with the GLEAM and GLDAS products, concluding the
uncertainty of monthly ET of 14.7 mm/m in the Yellow River basin (ID: 24) and 35.9 mm/m in the Pearl River basin (ID: 48),
nearly half of the estimates in our study, i.e., 27.0 and 71.7 mm/m in these basins, respectively, primarily due to different
datasets and methods used. We note these regional studies generally used observed and typically single-source water
components data like P and R, which can be the reason for the differences with our results based on multi-source data-based
calculations. Moreover, the difference in study region boundaries, data processing algorithms, calculation scheme of the
terrestrial water storage change, and time period may reflect the disparities in the estimates (Rodell et al., 2004).

A few global analyses can also provide an important reference for the ET-WB developed in our study. Specifically,

Zeng et al. (2012) collected in-situ runoff, precipitation, and GRACE data to estimate ET over 59 major river basins during
2003-2009, highlighting the fact that ΔS cannot be neglected in the water balance computations. This finding implies the
importance of including GRACE TWSA (ΔS) in the water balance closure at basin scales. Ramillien et al. (2006) applied the
GRACE samplings, GPCC precipitation, and modeled runoff to estimate ET time series over 16 drainage basins of the world,
in which the extreme errors (1.8 mm/day, 50% relative error) as expected by the accuracy of model runoff in the Amazon (ID:
1) River basin, is emphasized to influence the regional ET estimations. This well corresponds to the high uncertainty estimates
of P, R, and therefore ET-WB in both long-term mean and annual trend levels of our study. Similar to the examinations of
long-term mean and annual trends in our study, a previous global evaluation of water balance ET estimates against nine ET
products over 35 basins points out that water balance ET can reasonably estimate the annual means (especially in dry zones
with relatively lower uncertainty) but substantially underestimated the inter-annual variability in terms of annual trends and
mean annual standard deviation (Liu et al., 2016). Furthermore, the comprehensive uncertainty analysis for ET products from



four LSMs in NLDAS, two remote-sensing-based products including MODIS and AVHRR, and water balance estimations
show the highest uncertainty in the latter (20-30 mm/m) over the different climatic regions (from humid to arid) in South
Central United States (Long et al., 2014). The finding confirms the pattern of obviously higher uncertainty in ET-WB than
auxiliary ET products in several arid basins in Western United States in our study. A recently published global ET product
based on the three-temperature model used the water balance ET in 34 catchments worldwide as a benchmarking product,
revealing the RB mostly ranging from -25 to 25% on the annual scale, with the underestimation of water balance ET in high
latitudes (Yu et al., 2022). The comparisons are quite relevant to the results of ET-WB, which also underestimates ET in East
Russia and Northern North America by comparing with, for example, GLEAM and MODIS products. Overall, the results of
our proposed ET-WB datasets are consistent with previous regional and global studies, more importantly, cover the most recent
time periods, and provide observational constraints to the global and regional ET leveraging huge datasets of water balance
components.

**5.2 Implications, limitations, and future outlook**

The production of ET-WB ensemble datasets can benefit the future hydrological community in various ways. First of all, the
ET-WB can provide valuable information for the regional ET variations, greatly enriching the existing ET datasets consisting
of the remote-sensing-based (e.g., MODIS), LSM-predicted (e.g., GLDAS), GHM-predicted (e.g., WGHM), observation-
driven (e.g., FLUXCOM), in-situ-based (e.g., eddy tower observations) and other diagnostic datasets (e.g., GLEAM) as well
as the synthetic datasets. Given the non-ignorable differences among the existing ET datasets and an independent mass
conservation-based ET-WB, it can not only help to benchmark other datasets/models of ET but will also contribute to the
validation and calibration of hydrological models across scales. This is particularly useful for poorly gauged regions like the
Qinghai-Tibetan Plateau, African river basins, and high-latitudes cold regions, where the installation and maintenance of the
field observation network are quite challenging (Li et al., 2019). In addition, the ET-WB product will provide additional
information for evaluating water balance closure on the basin and global scales (Lehmann et al., 2022). The ET-WB dataset
that generates ET based on the terrestrial water balance is also dedicated to evaluating other water balance components like R
by combining them with the available hydrological records (e.g., P) regionally or globally (Syed et al., 2010; Chandanpurkar
et al., 2017). Finally, the ET-WB product is conducive to detecting human footprints in the regional water cycle. For example,
Pan et al. (2017) combined the water balance estimations of actual ET and the modeling results without consideration of human
activities to estimate human-induced ET in a highly developed region of China (Haihe River basin), implying a 12% increase
of ET due to human activities such as irrigation. Strong influences of anthropogenic changes to the region ET were also
reported in the Colorado River basin of western United States (Castle et al., 2016). Overall, the developed ET-WB has the
potential to support multi-discipline applications in hydrology and climate fields.
However, the ET-WB also suffers from a few limitations mainly related to the uncertainty, selection, and assumptions
of datasets involved in water balance computations. As shown in comparison with other ET datasets and the uncertainty
analysis, propagated uncertainty from different variables like ΔS and P can greatly influence the quality of ET estimations. For



example, the relatively higher uncertainty of GRACE signals in smaller basins increases after the derivation of ΔS subsequently
alters the estimations of ET. Biases in P over humid zones can also play an important role in the performance of regional ET.
In terms of R, since only one of the 29 subsets is from the in-situ discharge and mostly are provided by the observation-driven
machine learning G-RUN ENSEMBLE dataset with varying forcings, the ET estimations for the basins without in-situ
observations might be biased. Further, the G-RUN ENSEMBLE, as a gridded runoff rate product purely forced by
meteorological data, does not physically account for human activities (e.g., dam management) into consideration. Such
simplicities might overestimate or underestimate the actual runoff for the basins with significant human intervention, with the
underlying assumption that the water loss in river channels can be neglected to convert runoff into river streamflow on a
monthly scale. Overall, the inherent uncertainties in multiple water cycle components (P, ΔS, and R) can propagate to the ET-
WB product.

To overcome the multisource uncertainties, several suggestions for future use and improvements are provided as

follows: (1) appropriate consideration of human disturbances such as water diversion in water balance estimates of ET should
be highlighted in specific regions (e.g., the South-to-North Water Diversion Project across South and North China); (2)
considering the significant role of the forcing data in determining the accuracy of ET-WB, careful justification of different
inputs (e.g., P) that have better performance for the regions of interest should be performed in combination of regional in-situ
observations; (3) future efforts should incorporate in-situ ET observations from regional eddy covariance towers with
calibration, assimilation, and correction procedures to improve further the accuracy of ET-WB (Billah et al., 2015); (4)
integrated ET products that consider a hybrid approach to integrate strengths of different categories of data, including ET-WB
and satellite products, are worthy of being proposed to further constrain the uncertainties in regional ET (Long et al., 2014).
**6 Data availability**
All the datasets used in our study are publically available online and have been introduced in the Data section. The ET-WB
dataset is also publicly available in the Version 7.3 MAT-files (Xiong et al., 2023) and can be freely downloaded on the Zenodo
platform (doi:10.5281/zenodo.7314920).
**7 Conclusions**
In the current study, a global monthly ET product (named ET-WB) over 168 river basins that account for ~60% of the Earth's
land area except for Greenland and Antarctic ice sheets and global land during May 2002-December 2021, is developed based
on the water balance equation employing 23 precipitation, 29 runoff, and 7 ΔS datasets from satellite products, in-situ
measurements, reanalysis, and hydrological simulations. The performance of ET-WB has been evaluated against four auxiliary
global ET datasets comprising the GLEAM, FLUXCOM, MODIS, and WGHM at various time scales based on different
statistical metrics (i.e., CC, NSE, RMSE, and RB). The long-term mean and annual trend of ET-WB and above ET products





are also assessed. Uncertainty of ET-WB is quantified by propagating the errors in different water components, and its
relationships with basin size, climate aridity, and human irrigation are also investigated.
The seasonal cycles of the ET-WB ensemble, mainly dominated by precipitation, generally agree with multiple ET
global products despite the overestimations/underestimations in specific months compared with the median ET-WB results.
Inter-annual variability of global land ET-WB presents a gradual increase from 2003 to 2010 and a subsequent decrease during
2010-2015, followed by a sharper reduction in the remaining years due to the varying P, similar to other ET products. However,
the increase of P during 2015-2021 does not translate to the enhancement of ET because of the overestimated GloFAS
reanalysis and the limited data availability (e.g., G-RUN ENSEMBLE) in the period. Multiple statistical metrics show
reasonably good accuracy of ET-WB, with most river basins having RB between -20% and 20% on a monthly scale. The
performance improves on an annual scale but with strong spatial heterogeneity among different basins.
The long-term mean annual ET estimates from ET-WB are concentrated within the range of 500-600 mm/yr among
ensemble members with the median estimates of 549 mm/yr for global land, comparable to the result from GLEAM (543
mm/yr), MODIS (569 mm/yr), and WGHM (534 mm/yr). The relatively higher value from FLUXCOM (663 mm/yr) can be
attributed to the non-consideration of the unvegetated area. Regarding annual trends, the 'dry gets drier and wet gets wetter'
paradigm can be inferred from ET-WB, which generally exaggerates the prevailing increasing/decreasing ET in basins. On a
global scale, the median value of trend estimates from ET-WB ensemble members is 1 $mm/yr^2$, close to the results from
GLEAM (0.8 $mm/yr^2$) and WGHM (0.8 $mm/yr^2$). However, both FLUXCOM and MODIS report small negative values of -
0.3 and -0.1 $mm/yr^2$, respectively, still within the ET-WB ensemble spread range.
The uncertainty of ET-WB that roughly fluctuates below 15 mm/m (RMS: 9.7 mm/m) is primarily controlled by that
of P (RMS: 8.3 mm/m), which is relatively higher than the auxiliary ET products (RMS: 5.3 mm/m) over global land. The
inversely proportional relationship between uncertainty in ΔS and basin size governs the increasing uncertainty of ET-WB
along with the enhancement of basin area. Other variables like P and R present relative dependence upon the basin's aridity,
where they are inclined to have higher uncertainty in more humid regions with higher AI. Moreover, the significant irrigation
equipped for groundwater resources can lead to significant short-term, and long-term variations of, for example, ΔS and R,
which is the case of some basins in North China (e.g., Haihe River basin (ID: 67)) and North India (e.g., Indus River basin (ID:
27)). The uncertainty in ET-WB can be further intensified for the small wet basins with significant human disturbance, so
caution should be taken when drawing the scientific conclusions using ET-WB over those regions.
**Author contributions**
Jinghua Xiong contributed to the data processing. Jinghua Xiong and Abhishek conducted the research and wrote the original
draft and revised it. Gionata Ghiggi and Yun Pan contributed to the conceptual design and review of the manuscript. Shenglian
Guo contributed to the funding acquisition and project administration. All co-authors reviewed and revised the manuscript.



**Competing interests**
The authors declare that they have no conflict of interest.
**Acknowledgments**
We thank Dr. Shuang Yi from the University of Chinese Academy of Sciences for providing the reconstructed GRACE data.
**Financial support**
Our study was financially supported by the National Natural Science Foundation of China (U20A20317) and the National Key
Research and Development Program of China (2022YFC3202801).

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
