# Peer review of "ET-WB: water balance-based estimations of terrestrial evaporation"

_Earth System Science Data, 2023_

## Author Comment (AC2)

**Reply to Reviewers' comments (Reviewer#2)**

**Legend**
Reviewers' comments
Authors' responses
Direct quotes from the revised manuscript

We thank the reviewer for his/her time in reading our manuscript and detailed comments on our manuscript. Point-by-point replies to the comments or suggestions made can be found below.

**Reviewer #2:** The paper addresses a critical aspect of water cycle analysis by developing a water-balance based global ET dataset. It assesses uncertainties of multiple open access datasets on precipitation, runoff and storage change and performs water balance analysis on their combinations to derive ET by major river basin. This ET data is then compared with four auxiliary datasets developed through different methods.

Response: We thank the reviewer for his/her time in reviewing our manuscript, providing comprehensive suggestions for improvement, and recognizing the potential for publication of the ET-WB dataset.
Revisions for texts and datasets have been made in the new version, as suggested. Please find our specific response to your comments below.

The input data, methods and results are described in details and presented in clear and well-structured way. The dataset is only provided in MATLAB proprietary format, which may limit its accessibility.

Response: We thank the reviewer for the valuable suggestion. We have re-processed the ET-WB dataset in spatially distributed formats like NetCDF (nc) and ArcGIS shapefile (shp) formats and provide the geospatial database of global river basins and global land considered in our study. So now there are a total of three formats available for ET-WB (i.e., nc, shp, and mat). The updated link for this data is: https://zenodo.org/record/8151534

**Specific comments:**

(1) Data redundancy and circularity implications could be discussed. For example, runoff models are likely calibrated on GRDC archive, which is also directly used as input data;

Response: We thank you very much for the suggestion. We have added a discussion about the redundancy and circularity implications in the new version as below:
Another potential source of uncertainty is the redundant use of specific variables among datasets. For example, the G-RUN ENSEMBLE dataset also applies the station discharge from the GRDC archive. Such circular application of GRDC discharge in comparison with G-RUN might overestimate its true performance, and a similar situation may occur in the resulting ET-WB as the in-situ discharge also serves as its direct input.

(2) While results are also presented geographically in the paper, the output dataset is only distributed as matrix in mat file. Providing access to the dataset in different formats, including spatially distributed formats, would probably increase its usability;

Response: As suggested, we have re-distributed the ET-WB dataset into three different formats, including the widely used NetCDF format (nc), ArcGIS-friendly shapefile format (shp), and the

original matrix format (Mat), of which the former two are specially designed for geographical applications. The updated link for ET-WB is: https://zenodo.org/record/8151534

(3) The temporal extent of datasets used in the study (Table 1) could be explained. For example, the auxiliary ET datasets are used over a shorter temporal span without a clear justification;

Response: We have added explanations for the inconsistent spatial and temporal resolutions among datasets (e.g., water balance factors and auxiliary ET products) in the Data Section and the caption of Table 1 as below. Please note that Table 1 has been moved into the supplementary file according to the *Specific Comment #7* from Reviewer 1.
*Table 1 (Table S3 in the revised manuscript) Caption:*
Various datasets with different spatial resolutions are processed as area-averaged values over 168 river basins worldwide and global land for spatial consistency. For convenience in calculations, all the constituent water balance variables for calculating ET-WB are necessarily extended to the study period 2003-2021, with the missing months (temporally not covered) replaced with NaN values. However, only the overlapping period between ET-WB and four auxiliary ET products are extracted for comparisons, i.e., 02-14 for MODIS, 02-15 for FLUXCOM, 02-21 for GLEAM, and 02-16 for WGHM, respectively.

(4) Line 252, CHIRPS data is a combination of remote sensing and stations data, should it be considered as a combined product?

Response: Thank you very much for the comment. As indicated by the reviewer, the CHIRPS algorithm incorporates remote sensing and stations data. However, it is basically built on the long-lasting infrared satellite Cold Cloud Duration (CCD) observations (Funk et al., 2015). The stations data are only merged for post-processing, including bias-correction and spatial interpolation. Thereby, the CHIRPS precipitation database is classified as a satellite-based dataset in our study, unlike other combined products that equally apply the multi-source precipitation data from reanalysis, in-situ stations, and satellites (e.g., MSWEP).
Reference:
Funk, C., Peterson, P., Landsfeld, M. et al. The climate hazards infrared precipitation with stations-a new environmental record for monitoring extremes. Sci Data 2, 150066 (2015). https://doi.org/10.1038/sdata.2015.66

(5) Line 35, typo, "for" should read "four";

Response: Corrected in the new version.

(6) Line 96-97, consider rewording for better clarity.

Response: We have re-organized this sentence as suggested:
Previous studies used the water balance approach that either relies on single datasets (e.g., precipitation and/or runoff) (Gibson et al., 2019; Liu et al., 2016) or focuses on the regional scales (Castle et al., 2016; Pascolini-Campbell et al., 2020; Rodell et al., 2004; 2011; Swann et al., 2017; Wan et al., 2015).

---

## Author Response (AR1)

**Reply to Reviewers' comments (Reviewer#1)**

**Legend**

Reviewers' comments

Authors' responses

Direct quotes from the revised manuscript

We thank the reviewer for his/her time in reading our manuscript and detailed comments on our manuscript. Point-by-point replies to the comments or suggestions made can be found below.

**Reviewer #1:** This research provides a large ensemble of water balance ET estimations using existing global datasets. Sound comparisons with mainstream ET products have been reported. Uncertainty assessment is also performed on different scales. The manuscript is within the scope of ESSD journal. Minor revisions need to be considered before publication. Please find details below:

Response: We thank the reviewer for his/her time in reviewing our manuscript, providing comprehensive suggestions for improvement, and recognizing the potential for publication of the ET-WB dataset. Revisions for texts and figures have been made in the new version, as suggested. Please find our specific response to your comments below.

(1) ET-WB incorporates both the natural ET from various sources and human-induced part, e.g., irrigation. What about the products used for validation, particularly for GHM such as WaterGap? Can human activity be a significant reason causing difference between ET-WB and them?

Response: As rightly indicated by the reviewer, ET-WB includes the influences from both natural and anthropogenic systems because it is calculated using the physics-based water mass balance method forced by either pure observations (e.g., satellite products) or realistic simulations (e.g., GloFAS), both have incorporated natural and human-induced parts.

Please find the details about the auxiliary ET products below,
*MODIS, GLEAM, and FLUXCOM products*: In terms of the three auxiliary global ET databases (MODIS, GLEAM, FLUXCOM) for comparison, the governing mechanisms are divergent but consistent on the line of representing combined impacts from natural and human activities. For direct observational ET data from space (MODIS) and ground (FLUXCOM), the sources of ET are not partitioned so they also incorporate the human impacts because the actual earth surface states (e.g., albedo) and flux (e.g., water vapour exchange) are measured. GLEAM, which uses satellite data to estimate potential ET and then convert it to actual ET by a stress module (assimilated with soil moisture observations), also shares a similar condition.
*WGHM case*: The standard run of WGHM that is used in our study models major human activities like irrigation, dam construction (including their commissioning years), reservoir management, and regulated lakes, and therefore the resulting ET also includes the influences of these human activities (Müller Schmied et al., 2021).

Therefore, all the selected ET products represent the total ET, and the observed differences with ET-WB primarily reflect the bias in modelling, measuring, and/or processing. Despite this, slight differences in the representation of human activities in various datasets might have led to minor discrepancies in our results and may be studied in the future as and when better datasets/models are available.

Reference:
Müller Schmied, H., Cáceres, D., Eisner, S., Flörke, M., Herbert, C., Niemann, C., Peiris, T. A., Popat, E., Portmann, F. T., Reinecke, R., Schumacher, M., Shadkam, S., Telteu, C.-E., Trautmann, T., and Döll, P.: The global water resources and use model WaterGAP v2.2d: model description and evaluation, Geosci. Model Dev., 14, 1037–1079, doi:10.5194/gmd-14-1037-2021, 2021.

(2) The authors evaluate ET-WB on several time scales longer than a month. What about the performance on the sub-monthly scale? This point should either be supplied or discussed.

Response: Thanks for bringing up this point. We see potential in the sub-monthly scale evaluation of the water balance in essence. However, there are two major challenges that continue to make such analyses and/or subsequent interpretations ambiguous.
   1) *Coarse resolution of the available datasets:* Limited by the availability of directly observed input datasets (i.e., P, R, and $\Delta$S) on a finer scale, ET-WB is evaluated on the monthly timescale and evaluated only on timescales longer than a month (i.e., month, year, and multi-year).
   2) *Outweighed uncertainties at finer scales*: Although some of the water balance variables, for example, GRACE-based $\Delta$S, have been interpolated on a daily scale (Kvas et al., 2019), their performance over the native coarser resolution is much debated. Similarly, more extreme values of, for example, precipitation, will likely be observed on shorter time scales with higher uncertainty (Kim et al., 2020; Tabari, 2020). Therefore, sub-monthly (e.g., daily) calculations of ET-WB tend to pose challenges in confident interpretations, especially in a quasi-global domain.
Nonetheless, timely updates to our ET-WB dataset would be integrated when appropriate datasets (e.g., daily data of constituent water balance components) can be acquired. We have added this point in the Discussion Section of the revised version (see Lines 802-804):
Finally, due mainly to the availability of most input data, ET-WB covers a specific period (2002-2021) at a relatively coarse timescale (monthly). Higher frequency and longer duration are our future objectives when more data can be accessed.

References
Kvas, A., Behzadpour, S., Ellmer, M., Klinger, B., Strasser, S., Zehentner, N., & Mayer-Gürr, T. (2019). ITSG-Grace2018: Overview and evaluation of a new GRACE-only gravity field time series. Journal of Geophysical Research: Solid Earth, 124. https://doi.org/10.1029/2019JB017415
Kim, S., Eghdamirad, S., Sharma, A., &Kim, J. H. (2020). Quantification of uncertainty in projections of extreme daily precipitation. Earth and Space Science,7, e2019EA001052. https://doi.org/10.1029/2019EA001052
Tabari, H. Climate change impact on flood and extreme precipitation increases with water availability. Sci Rep 10, 13768 (2020). https://doi.org/10.1038/s41598-020-70816-2

(3) The authors explain the spatial pattern of uncertainty in ET-WB by individually presenting the uncertainty in different variables. However, the temporal changes lack necessary justifications, e.g., the decrease of auxiliary ET products in 2016 according to Fig. 9f.

Response: The abrupt decrease of uncertainty in auxiliary ET products after 2015 is due to the limited time coverage of FLUXCOM and MODIS products. Since the number of ET products used in the uncertainty calculations based on the standard deviation method has decreased from four to two, the RMS value of uncertainty has abruptly dropped from ~5 to ~2 mm/m. This decline does not necessarily mean that the uncertainty of ET products has decreased since 2015, while more likely a mathematical result of the lesser ensemble members. We have added explicit descriptions for such behaviour in the revised manuscript as follows (see Lines 691-693):
Similarly, the abrupt decrease of uncertainty in auxiliary ET products after 2015 is due to the limited time coverage of FLUXCOM and MODIS products, with an RMS of 5.3 mm/m over the whole period. They are not involved in the calculation of uncertainty based on the inter-member deviation since the year 2016.

(4) It would be good if the authors could add the uncertainty range of the selected auxiliary ET products in the comparisons with ET-WB in terms of the long-term mean and annual trends.

Response: As suggested, we have added the change range of uncertainties, denoted as the standard deviation across the ensemble members, citing the lack of available uncertainties in the individual

ET products (Long et al., 2014), in both the long-term mean and annual trends in four ET products in Figs. 7-8 of the revised version.

Reference:
Long, D., Longuevergne, L., and Scanlon, B. R.: Uncertainty in evapotranspiration from land surface modelling, remote sensing, and GRACE satellites, Water Resour. Res., 50, 1131–1151, doi:10.1002/2013WR014581, 2014.

(5) Fig. 1. The pie chart representing the irrigation percentage rate should be enlarged.

Response: As suggested, we have increased the size of the pie chart for better readability in Fig.1 of the new version.

(6) You may indicate the explicit number of datasets used in your workflow in Fig.2.

Response: We have added the exact number of applied datasets adjacent to each variable in Fig.2 of the new version as below. We have also modified the figure caption accordingly.

[Figure]

**Figure 2: Flowchart and the characteristics of the data sets in the study. Please see the data section for detailed descriptions of the various datasets. Numbers in the parentheses denote the number of the particular datasets used in our study.**

(7) Table 1 may go to supplementary file as (1) it occupies too much space, (2) it is somehow duplicate with Fig.2.

Response: Thank you for the suggestion. We have moved it to the supplementary file (Table S3).

(8) For colored shading plots like Figs. 3 and 5, the authors should indicate the meaning of the central red line in the legend.

Response: For these coloured shading plots, the shading area shows the spread range among different datasets and the central solid line meaning the ensemble median value. We have added explanations in the figure captions.

(9) Line 702-704: 'The human-induced inordinate fluctuations of water balance (e.g., through reservoir management, groundwater extraction) can influence the quality of ET-WB by impacting the accuracy of the specific forcing variable (e.g., R).' The statement needs to be re-organized for better comprehension.

Response: We have re-organized this statement as follows in the revised manuscript (see Lines 706-708):
The human-induced inordinate fluctuations can influence the water balance and subsequently the quality of ET-WB by impacting the accuracy of the specific forcing variable (e.g., impact R through reservoir management).

(10) The authors should either direct the authors to Table S2 or explain the meaning of the rank of basin ID in Fig. 10.

Response: We have revised the caption of Fig. 10 to (a) explain the meaning of the basin ID and (b) direct the readers to Table S2 for salient features of the river basins.

**Legend**

Reviewers' comments

Authors' responses

Direct quotes from the revised manuscript

We thank the reviewer for his/her time in reading our manuscript and detailed comments on our manuscript. Point-by-point replies to the comments or suggestions made can be found below.

**Reviewer #2:** The paper addresses a critical aspect of water cycle analysis by developing a water-balance based global ET dataset. It assesses uncertainties of multiple open access datasets on precipitation, runoff and storage change and performs water balance analysis on their combinations to derive ET by major river basin. This ET data is then compared with four auxiliary datasets developed through different methods.

Response: We thank the reviewer for his/her time in reviewing our manuscript, providing comprehensive suggestions for improvement, and recognizing the potential for publication of the ET-WB dataset.
Revisions for texts and datasets have been made in the new version, as suggested. Please find our specific response to your comments below.

The input data, methods and results are described in details and presented in clear and well structured way. The dataset is only provided in MATLAB proprietary format, which may limit its accessibility.

Response: We thank the reviewer for the valuable suggestion. We have re-processed the ET-WB dataset in spatially distributed formats like NetCDF and ArcGIS shapefile formats and provide the geospatial database of global river basins and global land considered in our study. So now there are a total of three formats available for ET-WB (i.e., nc, shp, and mat). The updated link for this data is: https://zenodo.org/record/8151534

**Specific comments:**

(1) Data redundancy and circularity implications could be discussed. For example, runoff models are likely calibrated on GRDC archive, which is also directly used as input data;

Response: We thank you very much for the suggestion. Since the GRUN model is calibrated only with a basin smaller than 2500 km$^2$, it is *not* calibrated with GRDC stations used for validation at the basin scale of the ET-WB comparison in our study (Ghiggi et al., 2021). However, LORA and the other GHMs are instead known to be partially calibrated with GRDC stations (Hobeichi et al., 2019), so the review concern applies mainly to the classical hydrological models rather than to the GRUN data. Nonetheless, we have added a discussion about the redundancy and circularity implications in the new version as below (see Lines 800-801):
Another potential source of uncertainty may arise from the redundant and circulatory use of specific variables (e.g., in-situ runoff data used in our calculations are also used for partially calibrating the GHM and LORA datasets) in the generation of ET-WB.
(2) While results are also presented geographically in the paper, the output dataset is only distributed as matrix in mat file. Providing access to the dataset in different formats, including spatially distributed formats, would probably increase its usability;

Response: As suggested, we have re-distributed the ET-WB dataset into three different formats,

including the widely used NetCDF format, ArcGIS-friendly shapefile format (shp), and the original matrix format (Mat), of which the former two are specially designed for geographical applications. The updated link for ET-WB is: https://zenodo.org/record/8151534

(3) The temporal extent of datasets used in the study (Table 1) could be explained. For example the auxilliary ET datasets are used over a shorter temporal span without a clear justification;

Response: We have added explanations for the inconsistent spatial and temporal resolutions among datasets (e.g., water balance factors and auxiliary ET products) in the Data Section and the caption of Table 1 as below. Please note that Table 1 has been moved into the supplementary file according to the *Specific Comment #7* from Reviewer 1.
*Table 1 (Table S3 of the revised manuscript) Caption:*
Various datasets with different spatial resolutions are processed as area-averaged values over 168 river basins worldwide and global land for spatial consistency. For convenience in calculations, all the constituent water balance variables for calculating ET-WB are necessarily extended to the study period 2003-2021, with the missing months (temporally not covered) replaced with NaN values. However, only the overlapping period between ET-WB and four auxiliary ET products are extracted for comparisons, i.e., 2002-2014 for MODIS, 2002-2015 for FLUXCOM, 2002-2021 for GLEAM, and 2002-2016 for WGHM, respectively.

(4) Line 252, CHIRPS data is a combination of remote sensing and stations data, should it be considered as a combined product?

Response: Thank you very much for the comment. As indicated by the reviewer, the CHIRPS algorithm incorporates remote sensing and station data. However, it is basically built on the long-lasting infrared satellite Cold Cloud Duration (CCD) observations (Funk et al., 2015). The station data are only merged for post-processing, including bias-correction and spatial interpolation. Thereby, the CHIRPS precipitation database is classified as a satellite-based dataset in our study, unlike other combined products that equally apply the multi-source precipitation data from reanalysis, in-situ stations, and satellites (e.g., MSWEP).
Reference:
Funk, C., Peterson, P., Landsfeld, M. et al. The climate hazards infrared precipitation with stations-a new environmental record for monitoring extremes. Sci Data 2, 150066 (2015). https://doi.org/10.1038/sdata.2015.66

(5) Line 35, typo, "for" should read "four";

Response: Corrected in the new version.

(6) Line 96-97, consider rewording for better clarity.

Response: We have re-organized this sentence as suggested (see Lines 97-99):
Previous studies used the water balance approach that either relies on single constituent datasets (e.g., precipitation and/or runoff) (Gibson et al., 2019; Liu et al., 2016) or focuses on the regional scales (Castle et al., 2016; Pascolini-Campbell et al., 2020; Rodell et al., 2004; 2011; Swann et al., 2017; Wan et al., 2015).